# Edible Coatings for Fresh Fruits: Functional Roles, Optimization Strategies, and Analytical Perspectives

**DOI:** 10.3390/plants15010132

**Published:** 2026-01-02

**Authors:** Siphumle Owen Jama, Robert Lufu, Umezuruike Linus Opara, Elke Crouch, Alemayehu Ambaw Tsige

**Affiliations:** 1Packaging and Cold Chain Research Group, Africa Institute for Postharvest Technology, Department of Horticultural Science, Faculty of AgriSciences, Stellenbosch University, Stellenbosch 7602, South Africa; jamas@sun.ac.za (S.O.J.); rlufu@sun.ac.za (R.L.); opara@sun.ac.za (U.L.O.); elke@sun.ac.za (E.C.); 2United Nations Educational, Scientific and Cultural Organization (UNESCO) International Centre for Biotechnology, Nsukka 410001, Nigeria

**Keywords:** coatings, biopolymers, shelf-life, sustainability, loss-reduction, polysaccharides, packaging

## Abstract

Fresh fruits are inherently prone to postharvest deterioration due to loss of moisture, respiration, mechanical damage, and microbial decay, making quality preservation a persistent challenge across fresh fruit supply chains. While conventional plastic packaging offers barrier protection and cost-efficiency, its environmental footprint, particularly poor biodegradability and increasing incidence of plastic waste necessitates a transition toward more sustainable alternatives. Among these, the use of edible coatings, primarily based on natural biopolymers, have emerged as a versatile strategy capable of modulating transpiration, gas exchange, microbial activity, and sensory quality while addressing environmental concerns. Unlike biodegradable plastic films, edible coatings directly interface with the fruit surface and offer multifunctional roles extending beyond passive protection. This review synthesizes recent advances in edible coatings for fresh fruits, with emphasis on material classes, functional performance, optimization strategies, and analytical evaluation methods. Key findings indicate that polysaccharide-based coatings provide adequate gas permeability but limited moisture resistance, while nanocomposite and multi-component systems enhance water-vapor barrier performance without compromising respiration compatibility. Incorporation of bioactive agents such as essential oils, nanoparticles, and plant extracts further extends shelf life through antimicrobial and antioxidant mechanisms, though formulation trade-offs and sensory constraints persist. The review also highlights critical limitations, including variability in barrier and mechanical properties, challenges in industrial-scale application, insufficient long-term validation under commercial cold-chain conditions, and regulatory uncertainty for active formulations. Future research priorities are identified, including mechanistic transport–physiology integration, standardized performance metrics, scalable application technologies, and life-cycle-informed material design. Addressing these gaps is essential for transitioning edible coatings from experimental sustainability concepts to robust, function-driven solutions for fresh-fruit preservation.

## 1. Introduction

Fresh fruits suffer postharvest losses of up to 50%, mainly due to moisture loss, decay, mechanical damage, aging, issues exacerbated by inadequate cold chain infrastructure in many regions [1]. The global fruit industry is under increasing pressure to balance quality preservation, consumer safety, and environmental sustainability across postharvest supply chains. Fresh fruits, being physiologically active and highly perishable, are highly vulnerable during handling, storage, and transportation.

Packaging plays a key and dynamic role in postharvest handling of fruit and other horticultural produce [2,3,4,5,6]. Its function include protecting fruit from mechanical damage during handling and bulk transport [2,7,8,9,10,11,12], reducing moisture loss [13,14,15], preventing contamination and helping to modify the gas composition around the produce [16]. Packaging also improves the marketability of fruit at the retail level. A variety of materials and designs are used, depending on the fruit type, storage needs, distribution logistics, and marketing conditions [17].

Traditionally, petroleum-based plastic packaging has been the dominant choice due to its excellent barrier properties, low cost, and scalability [18]. However, growing environmental concerns over plastic waste, along with regulatory changes and consumer demand for eco-conscious options, have spurred interest in eco-friendly alternatives [18,19]. Among these, edible coatings, which are thin, consumable films applied directly onto fruit surface, have emerged as a promising class of bio-based solution [20]. Composed of natural polymers such as polysaccharides, proteins, and lipids, these coatings can serve as semi-permeable barriers to moisture, gases, and solutes, thereby modulating respiration rates, reducing water loss, and extending shelf life [1]. Edible coatings can function not only as physical barriers but also as delivery systems for functional agents such as antimicrobials, antioxidants, and ripening inhibitors, thereby enhancing their protective efficacy [20,21,22]. When enriched with bioactive compounds such as plant-derived essential oils, these systems exhibit enhanced preservation [23,24].

Despite this potential, the application of edible coatings in the fresh fruit industry faces multiple critical challenges. Firstly, material compatibility is a major concern; the effectiveness of a coating depends on its interaction with the fruit’s surface morphology, wax layer, and respiration physiology, and these properties vary widely between species and even cultivars [25]. Secondly, coating functionality is limited by trade-offs among water barrier properties, gas permeability, and mechanical strength.

Another persistent issue affecting widespread use of edible coating is application scalability. While many edible coatings demonstrate efficacy under laboratory conditions, transitioning these systems to industrial-scale operations remains difficult due to challenges in uniform application, drying, and adhesion under variable humidity and temperature [26]. Moreover, regulatory ambiguity and lack of standardized approval procedures for edible coating formulations, particularly for coatings that include bioactive compounds, further hinder commercial adoption in global markets [27,28]. Additionally, low consumer acceptance poses a barrier, especially where coatings alter surface appearance, mechanical and surface properties, or perceived naturalness.

Furthermore, there is increasing evidence supporting the importance of developing holistic postharvest strategies. These strategies should integrate edible coatings with complementary cold chain postharvest technologies to fully enhance their potential. For instance, Valdés et al. [29] highlighted the synergy between edible coatings and controlled atmosphere (CA) storage, showing their combined effect in reducing respiration rate and microbial spoilage in strawberries. In this regard, the compatibility of coatings with packaging formats that maintain humidity, gas balance, and physical protection is vital to avoid coating degradation, cracking, or inefficacy. The integration of such coatings into a smart packaging context, which may involve simplified indicators or passive sensors for ripening and freshness, can bridge functionality with sustainability, supporting both quality maintenance and reduced plastic reliance [30].

Although the literature covers a wide range of biopolymer-based coating materials and application techniques, research remains fragmented. Most studies focus on short-term physicochemical changes, overlooking long-term interactions between coating composition, fruit metabolism, and storage environment. Furthermore, comparative evaluations across fruit types, coating systems, and storage scenarios are limited, and comprehensive lifecycle or techno-economic assessments are rare. Given the breadth of the edible coatings literature and the diversity of fruit systems, this review does not attempt to exhaustively catalog all coating materials or application outcomes reported to date. Instead, it adopts a selective and analytical approach, focusing on representative studies that illustrate key functional mechanisms, optimization trade-offs, and implementation challenges relevant to fresh fruit preservation. Depth is therefore emphasized in areas where coating performance intersects most strongly with fruit physiology, storage environment, and supply-chain integration, while peripheral or highly specialized topics are referenced more concisely. This scoped approach is intended to provide clarity, coherence, and critical insight, rather than a purely descriptive compilation of studies.

This article is presented as a narrative review, synthesizing recent and authoritative primary studies to provide an integrative overview of edible coatings for fresh fruit preservation, rather than as a systematic literature review.

## 2. Edible Coating Materials

Table 1 summarizes the edible coating materials consistently supported by current postharvest literature, focusing on their moisture resistance, gas permeability, and demonstrated performance on specific fruits. In the context of fresh-fruit preservation, edible coating materials can be broadly categorized into four major classes: polysaccharide-based, protein-based, lipid-based, and composite or nanostructured systems [1,23,24,25]. Polysaccharides—including pectin, alginate, and chitosan—represent the most extensively studied and widely applied class for whole fruits due to their inherent gas permeability, biocompatibility, and compatibility with fruit respiration. However, their hydrophilic nature limits moisture barrier performance when used as standalone coatings, particularly under high-humidity storage conditions. Protein- and lipid-based materials have also been explored as edible coatings, but their application in fresh fruits is more constrained. Protein-based coatings offer good film-forming ability and mechanical strength but are highly sensitive to moisture plasticization, while lipid-based coatings provide excellent moisture resistance at the expense of restricted gas exchange, which can induce anaerobic metabolism if not carefully controlled. These material-specific limitations have driven the development of composite and nanostructured systems that combine complementary functionalities.

A semi-quantitative heatmap (Figure 1) was developed to enable structured comparison of representative edible-coating systems across four functional dimensions: moisture resistance, gas-permeability compatibility, shelf-life extension potential, and physiological risk (horizontal axis). Each coating material or composite system (vertical axis) was assigned an ordinal score (1 = low, 2 = moderate, 3 = high) based on consistent experimental trends reported for water-vapor barrier performance, respiratory gas transport, storage-life outcomes, and physiological safety under typical postharvest conditions.

Single-component polysaccharide and protein coatings generally exhibit moderate moisture resistance and favorable gas-exchange compatibility, supporting low physiological risk but achieving only limited shelf-life extension unless formulation is optimized. In contrast, lipid-based coatings (e.g., carnauba/beeswax) score highly in moisture resistance but rank lower in gas-permeability compatibility and physiological safety, reflecting the documented tendency of dense wax layers to restrict O_2_ diffusion, elevate internal CO_2_ levels, and induce anaerobic metabolism and off-flavor development when coating thickness or loading is excessive [31]. Composite and nanostructured systems demonstrate more balanced performance across all four dimensions; for example, CNC-reinforced polysaccharide coating enhances moisture resistance and shelf-life extension while maintaining adequate gas permeability and physiological safety, consistent with the tortuous-diffusion effects reported by Pirozzi et al. [32]. Similarly, Pickering-nanoemulsion-based pectin coatings improve mechanical robustness and moisture-barrier efficiency without compromising respiratory compatibility or physiological stability [33]. This unified assessment framework highlights the inherent trade-offs associated with single-component systems and the balanced performance achieved by composite and nanostructured coatings. Together, these representations emphasize that coating effectiveness cannot be assessed on the basis of barrier properties alone but must be evaluated in relation to fruit-specific physiological requirements, including respiration rate, transpiration demand, and storage environment. Pirozzi et al. [32] reported that incorporating nanocellulose into edible-coating matrices enhances barrier properties and structural integrity, leading to improved firmness and reduced dehydration during storage. However, specific water-vapor transmission rate (WVTR) measurements were not provided. Similarly, de Oliveira Filho et al. [33] found that embedding Pickering nanoemulsions into pectin-based matrices increased mechanical strength and improved moisture barrier performance. González-Cuello et al. [34] developed a multi-component coating system combining bacterial cellulose, chitosan, and gellan gum, achieving up to 15 days of strawberry (*Fragaria* × *ananassa*) shelf-life extension while preserving antioxidant content and reducing enzymatic deterioration. In contrast, traditional polysaccharide coatings—such as pectin, alginate, and chitosan—offer only moderate moisture resistance due to their hydrophilic nature. These materials maintain appropriate gas permeability but struggle to limit water loss effectively. As noted by Liyanapathiranage et al. [35] and Miteluț et al. [36], pectin coatings can reduce weight loss in strawberries but still allow significant moisture permeation compared to nanostructured or composite formulations. Wigati et al. [37] report that cellulose nanocrystals (CNC) incorporation enhances barrier density without compromising oxygen and carbon-dioxide diffusion. These improved moisture barriers translate into better firmness retention and reduced dehydration, especially in high-respiration fruits such as strawberries, raspberries, cherries, peaches, and nectarines.

In addition to the primary biopolymer matrices, minor formulation additives such as plasticizers and surfactants play a decisive role in determining coating processability and physical performance. Plasticizers, including glycerol and sorbitol, are routinely incorporated into polysaccharide- and protein-based coatings to reduce brittleness and enhance flexibility; however, their concentration strongly influences water vapor permeability and mechanical stability due to increased polymer chain mobility [32,38]. Likewise, surfactants and emulsifiers are often required to improve the dispersion of hydrophobic components within aqueous coating formulations and to enhance wettability and adhesion on fruit surfaces, thereby promoting more uniform coating depositions during dipping or spraying [32]. These formulation-level modifications significantly affect coating microstructure and transport behavior and must therefore be considered integral to edible coating design rather than auxiliary adjustments.

The importance of tailoring edible-coating formulations to the physiological characteristics of individual fruits is clearly demonstrated across recent studies. Sun et al. [39] showed that modifying a pectin matrix with trans-cinnamaldehyde effectively addressed the rapid moisture loss and browning typical of rambutan, illustrating how additive selection must respond to a fruit’s specific deterioration pathways. Likewise, the performance of alginate coatings on fresh-cut papaya, as reported by Tabassum and Khan [40], reflects the coating’s compatibility with the high-respiration, enzyme-active nature of cut tissues—providing adequate gas exchange yet only moderate moisture protection. Chitosan-based coatings exhibit similar fruit-dependent behavior: while they support aerobic respiration across strawberries, apples (*Malus domestica*), and bananas (*Musa* spp.), they offer only limited resistance to the higher transpiration demands of these fruits [41,42]. Together, these findings underscore that no single polymer system performs optimally across all commodities; instead, coating composition, additives, and barrier properties must be strategically aligned with each fruit’s unique transpiration rate, respiration profile, and susceptibility to quality loss.

**Table 1 plants-15-00132-t001:** Summary of moisture-resistant edible coating materials tested in the fresh fruit industry, including their moisture barrier capacity, gas permeability behavior, and performance on fruit, as reported in recent studies.

Material/Composite	Moisture Resistance	Gas Permeability	Performance on Fruit	Corrected Citations
**Polysaccharides (pure)**				
Pectin (pure)	Moderate (limited by hydrophilicity)	Allows O_2_/CO_2_ exchange	Reduced weight loss in strawberries	[35,36]
Alginate (pure)	Moderate (hydrophilic nature)	Allows O_2_/CO_2_ exchange	Used widely; variable performance	[40]
Chitosan (pure)	Moderate (hydrophilic nature)	Allows O_2_/CO_2_ exchange	Applied across various fruits	[41,42]
**Protein-based coatings**				
Whey protein films	Low–moderate (highly sensitive to humidity and plasticizer content)	Low O_2_ permeability (good oxygen barrier; CO_2_ permeability not reported)	Good film-forming ability and mechanical integrity; moisture-barrier performance strongly formulation- and humidity-dependent	[38]
Zein-based coatings	Moderate (hydrophobic protein domains)	Reduced O_2_ permeability	Improved surface protection and firmness-retention potential; excessive coating thickness may restrict gas exchange	[43,44]
**Lipid-based coatings**				
Carnauba/Beeswax	High (hydrophobic wax layer strongly limits transpiration)	Low O_2_ permeability (coating thickness and formulation dependent)	Effective transpiration control and surface protection; excessive coating loadings may restrict gas exchange and promote anaerobic metabolism	[31]
**Composite & nanostructured systems**				
CNC-enhanced polysaccharides	Significantly improved (tortuous diffusion path)	Maintains O_2_/CO_2_ diffusion	Preserved firmness and reduced water loss in strawberries	[36,37,45]
Pectin + trans-cinnamaldehyde	High moisture retention	Maintains gas exchange	Reduced weight loss; quality retained in rambutan	[39]
Pickering nanoemulsions in pectin	Improved barrier and mechanical properties	Controlled permeability	Enhanced coating strength and moisture barrier	[33]
Bacterial cellulose + chitosan + gellan gum	High moisture retention (extended storage)	Allows respiration over 15 days	Extended shelf life and quality in strawberries	[34]

Beyond qualitative classification of coating materials, recent studies increasingly emphasize the need to quantify transport properties to enable rational coating design. Moisture-loss control is governed not only by polymer hydrophilicity but by coating thickness, microstructural tortuosity, and polymer–plasticizer interactions, which collectively determine effective water-vapor permeability (WVP) and diffusivity [36,38]. In protein-based systems, these trade-offs are particularly evident: plasticized whey-protein films exhibit good film-forming ability and mechanical integrity but remain highly sensitive to ambient humidity and plasticizer content, resulting in formulation-dependent moisture-barrier performance despite their low oxygen permeability [38]. Similarly, zein-based coatings, owing to their hydrophobic protein domains, reduce oxygen permeability and provide effective surface protection, yet excessive coating thickness can restrict gas exchange and pose risks of localized anaerobiosis if not carefully controlled [43,44].

Incorporation of nanocellulose or cellulose nanocrystals further enhances barrier performance by introducing a tortuous diffusion pathway that reduces effective vapor flux without proportionally restricting oxygen diffusion, a balance critical for maintaining aerobic respiration in climacteric fruits [37,38]. Composite systems that combine polysaccharides with lipophilic phases or nanoemulsions likewise modify the continuous-phase morphology, leading to non-linear reductions in WVP that cannot be predicted from bulk polymer properties alone [33,36]. In contrast, lipid-based coatings such as carnauba or beeswax form highly hydrophobic surface layers that strongly suppress transpiration but also exhibit low oxygen permeability; their performance is therefore strongly dependent on coating load and formulation, with excessive application potentially restricting gas exchange and promoting anaerobic metabolism [31].

From a physiological standpoint, excessive restriction of gas exchange has been linked to internal anaerobiosis, ethanol accumulation, and off-flavor development, particularly in high-respiration fruits such as strawberries and fresh-cut produce [40,41]. Collectively, these findings underscore that coating effectiveness must be evaluated as a coupled transport–physiology problem rather than as an isolated material property, highlighting the importance of integrating quantitative barrier metrics with fruit respiration characteristics when assessing edible coating performance.

## 3. Multifunctionality of Edible Coatings in the Fresh Fruits Supply Chain

The functional effects of edible coatings arise from their ability to modify mass transfer processes at the fruit–environment interface, as well as from the incorporation of active compounds that interact with microbial and biochemical pathways. Across material classes, coating performance is governed by differences in polymer chemistry, microstructure, and phase composition, which determine water vapor permeability, gas diffusivity, and interaction with fruit surface properties. Polysaccharide-based coatings typically form hydrophilic, semi-permeable films that maintain respiration but provide limited moisture resistance, whereas lipid-based coatings restrict transpiration more effectively but may impede oxygen diffusion if applied excessively. Protein-based coatings offer good film cohesion and mechanical integrity but are sensitive to moisture plasticization. Composite and nanostructured systems combine complementary mechanisms—such as tortuous diffusion pathways, phase separation, and controlled release of active agents—to achieve balanced moisture control, gas exchange compatibility, and antimicrobial efficacy. The following subsections examine these mechanisms in relation to specific functional outcomes, with Table 2, Table 3, Table 4 and Table 5 providing comparative evidence across coating systems and fruit types.

### 3.1. Moisture Management and Transpiration Control

Moisture management and transpiration control are essential for preserving postharvest fruit quality, as even moderate moisture loss can cause textural softening, shrinkage, and accelerated metabolic activity that sharply reduce marketable life [46,47,48]. Transpiration—driven by water-vapor diffusion through the cuticle and lenticels—is strongly affected by the type of produce, storage temperature, and relative humidity. For example, under ambient conditions (20–25 °C), tomatoes and leafy vegetables may lose 5–10% of their weight within 3–5 days, whereas apples and pears (*Pyrus communis*) held at 0–4 °C typically lose only 1–2% over two weeks [49,50]. Weight loss is generally quantified gravimetrically, and water-vapor transmission rate (WVTR) is expressed as grams of water lost per square meter per day (g/m^2^·day). Such losses impair turgidity, gloss, and perceived freshness, underscoring the need for effective moisture-control interventions during storage and distribution.

Plastic liners are commonly used in commercial packaging to maintain high relative humidity (90–95%) around bulk produce, reducing moisture loss by 40–70% under practical conditions [4]. These films—typically made of polyethylene or polypropylene—create a stable humid microclimate whose effectiveness depends on film thickness (20–50 µm), perforation density, and sealing integrity. Edible coatings, applied directly to the fruit surface, act as semi-permeable barriers that reduce WVTR while still allowing adequate gas exchange (oxygen and carbon dioxide). As summarized in Table 2, moisture-loss control by edible coatings is primarily governed by water vapor transport resistance, which varies systematically with coating material class and microstructural design. Polysaccharide-based coatings such as alginate, chitosan, or starch have been shown to reduce weight loss by 30–60% relative to uncoated produce [51,52], although their performance depends on coating thickness (10–80 µm), inherent hydrophilicity, and compatibility with produce respiration rates. Improved outcomes have been observed when these coatings are modified with functional additives—including essential oils, hydrophobic modifiers, and nano-reinforcers—which can strengthen film density, enhance water-binding capacity, and reduce permeability, as demonstrated in starch-based, composite, and nanoparticle-enhanced systems [49,50,51,52]. Cellulose-derived matrices and composite edible films further extend this potential by offering tunable microstructures and enhanced moisture-barrier integrity across a range of fruit commodities [50,53,54,55,56,57,58,59].

The performance of any moisture-control strategy is influenced by environmental conditions and the intrinsic properties of the material. Plastic liners provide durable and scalable protection, achieving up to 70% moisture-loss reduction even under fluctuating humidity [4], yet their non-biodegradable nature presents long-term environmental concerns. Edible coatings offer a biodegradable alternative but may vary in moisture-barrier effectiveness depending on formulation: plasticizers such as glycerol, for instance, can improve film flexibility but simultaneously increase WVTR by 10–15% [46]. Because storage temperature (0–25 °C) and RH (50–95%) are primary drivers of transpiration, coating composition and application protocols must be calibrated to match each fruit’s respiration profile, surface morphology, and susceptibility to dehydration. Table 2 illustrates the diversity of edible coating systems and reporting practices across fruit types, highlighting both quantified weight-loss reductions and trend-based outcomes where numerical values were not explicitly reported.

**Table 2 plants-15-00132-t002:** Moisture loss reduction by control strategies. Note: “Weight Loss Reduction (%)” values are derived from comparisons with uncoated controls when available. In cases where uncoated data is not directly provided, the percentage reflects reported or estimated reductions from baseline studies.

Produce	Coating System	Storage Conditions	Weight-Loss Reduction	Reference
Rambutan	Pectin + trans-cinnamaldehyde	10–15 °C, 12–15 days	≈35–45% WL reduction vs. control	[39]
Fresh-cut Papaya	Alginate coating	5 °C (MAP), 7–9 days	Significant WL reduction (trend reported; no % value)	[40]
Strawberries	BC + Chitosan + Gellan gum composite	4 °C, 12–15 days	WL significantly reduced (qualitative)	[34]
Strawberries	Chitosan + essential oils	4 °C, 10–15 days	WL reduction observed (no % reported)	[41]
Strawberries	CNC-reinforced chitosan	4 °C	Lower WL than pure chitosan (quantified trend)	[38]
Bananas	Nanoparticle-enriched chitosan	20–22 °C	WL reduction evident from graphs	[42]
Passion fruit	Cassava starch + ZnO nanoparticles	8–10 °C	Delayed WL; reduced slope of WL curves	[48]
Blueberries	EO-enriched edible coating	4 °C	WL reduction reported (graphical data)	[55]
Oranges	Carnauba wax	5–7 °C	WL reduction quantified	[60]

WL, weight loss; MAP, modified atmosphere packaging; BC, bacterial cellulose; CNC, cellulose nanocrystals; EO, essential oil; ZnO, zinc oxide.

### 3.2. Effect of Edible Coatings on Gas Exchange and Respiration

Table 3 synthesizes reported effects of edible coatings on gas exchange across multiple fruit types and coating material classes, illustrating how semi-permeable films regulate oxygen and carbon dioxide diffusion, thereby modulating respiration rate, ethylene production, and ripening-related quality attributes. These effects are strongly material-dependent, reflecting differences in polymer chemistry, microstructure, and effective gas diffusivity, as well as their interaction with fruit-specific metabolic demand. Edible coatings act as semi-permeable barriers to gases such as oxygen (O_2_), carbon dioxide (CO_2_), and ethylene (C_2_H_4_), thereby modifying the internal atmosphere of fresh fruits and vegetables (Table 3). This selective permeability helps regulate respiration rates, mitigate oxidative stress, and delay senescence, thereby extending postharvest shelf life and preserving quality attributes such as firmness, color, and flavor [46,50]. The ability of a coating to regulate gas exchange depends largely on its physicochemical composition and structural integrity. Polysaccharide-based coatings, such as those made from chitosan, alginate, or pectin, are hydrophilic and exhibit relatively high gas permeability, making them suitable for high-respiration fruits where excessive gas restriction could be harmful. In contrast, lipid-based or composite coatings incorporating waxes and fatty acids offer lower O_2_ permeability, which is beneficial for moderate-respiration fruits but may pose a risk of anaerobic conditions in produce requiring higher gas exchange.

Research has shown that chitosan coatings reinforced with cellulose nanocrystals can reduced O_2_ transmission while still allowing CO_2_ pass through. This selective gas control has been shown to delay ripening and maintain firmness in strawberries [51]. Similarly, edible films enriched with essential oils or hydrophobic compounds, such as sunflower wax, have provided improved control over CO_2_ accumulation and moisture loss in fruits like blueberries and grapes [52]. Such coatings effectively create a micro-modified atmosphere around produce, analogous to modified atmosphere packaging (MAP) but with enhanced sustainability benefits. However, excessively low O_2_ permeability may trigger anaerobic respiration, resulting in ethanol accumulation and undesirable flavor development. Thus, optimizing coating parameters, including thickness, composition, and application method, is essential to ensure that gas barrier performance align with the fruit respiration need [52]. Du et al. [53] demonstrated that by tuning the layer-by-layer assembly of chitosan and sodium alginate coatings, it is possible to match the gas barrier properties of the coating to the respiration characteristics of specific fruits. Their study showed that optimized coatings extended the shelf life of strawberries, tangerines, and bananas by 2, 4, and 4 days, respectively, by maintaining an appropriate modified atmosphere. Nonetheless, further studies are warranted to quantify O_2_ and CO_2_ transmission under dynamic storage conditions and to develop standardized approaches for coating customization based on produce type and storage duration.

**Table 3 plants-15-00132-t003:** Recent studies on edible coatings and gas exchange.

Coating System	Fruit	Observed Gas-Exchange Effect	Notable Findings	Reference
CNC–chitosan composite + oregano EO	Strawberry	Reduced O_2_ permeability; delayed ripening	Retained firmness and phenolics; WL reduced (10.8% vs. 37% control)	[51]
Chitosan–cellulose nanofibril (CNF) composite	Strawberry	Semi-permeable to O_2_ and CO_2_	Improved moisture retention, antioxidant preservation, firmness retention	[47]
Nanocellulose + myrtle EO	Strawberry	Controlled respiration rate	Minimized WL; maintained firmness and anthocyanins over 18 days	[54]
Chitosan + grape-seed EO	Strawberry, apple slices	Reduced O_2_/CO_2_ diffusion	Maintained vitamin C and polyphenols; reduced yeast/mold growth	[41]
CNC–chitosan composite (without EO)	Strawberry	Reduced O_2_ permeability	Delayed ripening; firmness and phenolic retention	[32]
Chitosan nanoparticle coating	Banana	Reduced respiration rate and ethylene evolution	Shelf life extended from 5 to 11 days	[42]
Self-healing alginate–Ca^2+^ matrix	Banana	Improved gas barrier	Reduced water loss and decay during storage	[57]
Aloe vera–gelatin + cinnamon EO nanoemulsion	Blueberry (*Vaccinium* spp.)	Reduced CO_2_/O_2_ exchange	Lower microbial spoilage; improved sensory scores	[55]
Chitosan + thyme EO composite	Blueberry	Reduced gas exchange and transpiration	Lower enzymatic browning; retained anthocyanin and vitamin C	[50]
Water-chestnut starch + rosemary EO nanoemulsion	Apple	Reduced respiration and ethylene release	Maintained firmness; reduced microbial activity	[56]
Carnauba wax (commercial)	Citrus	Reduced transpiration; limited O_2_ diffusion	Delayed senescence; reduced moisture loss	[60]
Zein-based protein coating	Tomato (*Solanum lycopersicum* L.)	Semi-permeable barrier; reduced transpiration and moderated respiration (inferred from ripening delay)	Reduced weight loss; delayed softening and color development; extended shelf life relative to uncoated fruit	[61]
Cassava starch + ZnO nanoparticles (LLDPE composite)	Passion fruit	Reduced CO_2_ accumulation and water loss	Extended shelf life; enhanced antibacterial properties	[48]

### 3.3. Microbial Control and Food Safety

Table 4 highlights how antimicrobial efficacy depends on both the intrinsic properties of the coating matrix and the mode of incorporation and release of active agents. Edible coatings have attracted considerable interest not only for their barrier properties but also as carriers for antimicrobial agents that enhance the microbial safety of fresh fruits. These coatings can inhibit microbial growth on fruit surfaces either through direct contact or by gradually releasing antimicrobial compounds into the fruit’s immediate environment. Among the most studied antimicrobial additives are essential oils, bacteriocins, and metal-based nanoparticles such as silver. Essential oils, particularly those derived from thyme, clove, and cinnamon, are rich in bioactive compounds like thymol, carvacrol, and cinnamaldehyde. These compounds exhibit broad-spectrum antimicrobial activity through mechanisms such as disruption of microbial cell membranes and interference with metabolic pathways. Sarengaowa et al. [58] demonstrated that a chitosan-based coating enriched with cinnamon oil significantly inhibited microbial growth, delayed browning, and maintained the quality of fresh-cut potatoes stored at 4 °C. Similarly, Vidyarthi et al. [59] found that coatings incorporating thyme and clove oils helped suppress spoilage and maintain the antioxidant activity in green chili, indicating similar potential for other fresh fruits.

Inorganic agents like silver nanoparticles (AgNPs) have also been incorporated in edible coating due to their potent and broad-spectrum antibacterial effects. Bizymis et al. [62] developed a multi-component coating based on chitosan, cellulose nanocrystals, β-cyclodextrin, and silver nanoparticles, which achieved over 96% reduction in Escherichia coli populations on cherries. Furthermore, this coating maintains fruit firmness and color during cold storage, while also decreasing oxygen and water vapor permeability. The integration of such coatings with cold storage protocols significantly improved microbial suppression compared to refrigeration alone, highlighting the value of synergies between physical and biochemical preservation methods. Nevertheless, while promising, the application of antimicrobial edible coatings is not without challenges. One major concern is the potential development of microbial resistance due to continuous exposure to sub-lethal concentrations of antimicrobials, particularly essential oils. Though resistance mechanisms like efflux pumps and membrane adaptation have been proposed, long-term data from fresh fruit systems remain limited [58]. Similarly, silver nanoparticle-based coatings raise questions regarding food safety, toxicity, and environmental persistence, as residues may remain on fruit surfaces post-application. These factors complicate regulatory approval. In certain jurisdictions, such as the United States or European Union, AgNP-containing coatings may be classified under pesticide legislation, requiring extensive safety evaluations before commercial deployment [62].

To mitigate these limitations, researchers advocate for the use of antimicrobial coatings in conjunction with traditional sanitation methods. For example, combining chitosan-thyme oil coatings with mild washing or UV-C treatment has shown improved efficacy in microbial control, without the need for high concentrations of active ingredients [59]. This integrative strategy may mitigate regulatory hurdles while still enhancing safety and shelf life. Despite extensive laboratory validation, significant research gaps persist. Most studies have focused on short-term storage under controlled conditions. There is a pressing need for long-term trials simulating real-world logistics involving fluctuating temperatures, handling stresses, and varying humidity levels. Moreover, little is known about the interaction between antimicrobial coatings and naturally occurring fruit microbiota, including beneficial epiphytic organisms. Addressing these gaps will require interdisciplinary research involving microbiology, food safety, postharvest physiology, and regulatory science. Scalable, safe, and effective antimicrobial coatings will only be realized when materials science converges with industrial practice and policy alignment.

**Table 4 plants-15-00132-t004:** Antimicrobial edible coatings in fresh fruits.

Coating System	Antimicrobial Agent	Produce	Effectiveness and Outcomes	Reference
Chitosan + clove essential oil (1%)	Eugenol, carvacrol	Strawberries	~50% WL reduction; delayed fungal growth over 12 days at 4 °C	[41]
Chitosan + cinnamon essential oil (1%)	Cinnamaldehyde	Fresh-cut potatoes	Significant reduction in *L. monocytogenes*; delayed browning at 4 °C	[58]
Chitosan–CNC–β-cyclodextrin + AgNPs	Silver nanoparticles	Cherries	>96% *E. coli* reduction; maintained firmness and color	[62]
Chitosan + ZnO nanoparticles	Zinc oxide nanoparticles	Passion fruit	Extended shelf life; enhanced microbial and physicochemical stability	[48]
Gum Arabic + methyl cellulose + thyme oil	Thymol, carvacrol (thyme oil)	Pomegranate arils	Reduced microbial growth relative to control; improved retention of phenolics and antioxidant capacity during 15 days of cold storage.	[23]
Gum Arabic + starch	Lemongrass essential oil	Pomegranate fruit	Reduced microbial growth (surface decay), Improved firmness retention, Better maintenance of anthocyanins and overall quality	[63]
Chitosan coatings with essential oils	Chitosan + essential oils (phenolic compounds)	Strawberries and Apples	Reduced weight loss and delayed quality deterioration during cold storage	[41]
Xanthan gum + Spirulina + pomegranate seed oil	Phenolics, fatty acids, antioxidant compounds	Mexican lime (*Citrus aurantifolia*)	Lowest WL; increased phenolics/flavonoids; inhibited PPO activity	[46]
Carnauba wax	Hydrophobic wax barrier	Citrus	Reduced surface fungal development; delayed senescence and quality deterioration during storage	[60]

EO = essential oil; WL = weight loss; PPO = polyphenol oxidase.

### 3.4. Nutritional and Sensory Quality Retention

Edible coatings play a crucial role in preserving the nutritional quality and sensory appeal of fresh fruits, maintaining key attributes such as vitamin C, phenolic content, color, flavor, and texture. As summarized in Table 5, preservation of nutritional and sensory quality emerges from the combined effects of barrier properties, oxidative protection, and avoidance of anaerobic or off-flavor development. Biopolymer-based films, especially those incorporating natural waxes, polysaccharides, proteins, or composites, have demonstrated effectiveness in safeguarding biochemical nutrients during cold storage, thereby extending shelf life and consumer acceptability. For example, carnauba wax coating applied to Moro oranges (*Citrus sinensis*) significantly reduced weight loss and helped maintain fruit firmness, anthocyanin levels, and vitamin C over an 80-day period. Although antioxidant levels declined over time, the preservation of firmness and color was deemed promising [60]. Polysaccharide films infused with pomegranate (*Punica granatum*) peel extract or Spirulina phenolics have similarly protected vitamin C and total phenolic content in mango, strawberry, and lime, with enhanced enzyme activity and delayed browning [46,64]. Texture and visual appearance remain core quality metrics. A sodium alginate–based coating applied to pineapple effectively preserved brightness and structural integrity without affecting sensory qualities [64]. Xanthan gum coatings with lemongrass oil on mandarin fruits preserved titratable acidity, soluble solids, vitamin C, and antioxidant levels while reducing weight loss and spoilage [65]. In apples, whey protein and zein-based coatings have been used to mitigate browning and maintain firmness and natural aroma over extended storage [66].

In a study by Siringul & Aminah [67], mango cubes coated with seaweed paste containing varying concentrations of Kappaphycus alvarezii and gum Arabic exhibited significantly reduced weight loss and retained firmness over 14 days of refrigerated storage. The coatings also maintained neutral sensory attributes, with no adverse effects on taste or aroma, and were well accepted by a trained panel. These findings suggest that *K. alvarezii*-based coatings can effectively create a micro-modified atmosphere around the fruit, reducing moisture loss and oxidative degradation while preserving texture. The study highlights the potential of seaweed coatings as a sustainable alternative to conventional packaging, especially for minimally processed tropical fruits. In a study by Dulta et al. [68], oranges coated with a layered formulation of 1% chitosan and 1.5% sodium alginate, supplemented with 0.5 g/L ZnO nanoparticles, exhibited markedly reduced mold growth, improved firmness, and higher retention of vitamin C over a 20-day refrigerated storage period. Similarly, carboxymethyl chitosan–gelatin coating maintained firmness and antioxidants in sweet cherries [69], while strong adhesion improved the retention of turmeric oil in chitosan films, enhancing antioxidant activity [70]. For black mulberries, coatings extended shelf life and preserved sensory traits [71], and alginate or chitosan coatings with avocado extract kept minimally processed apples fresh and appealing [72]. Overall, optimized formulations and adhesion are key to sustaining nutritional and sensory quality across diverse fruits.

Despite these benefits, sensory drawbacks occasionally emerge. Waxy coatings may impact mouthfeel or gloss, causing an unnatural sheen, and thick films may retain ethanol-like flavors due to restricted respiration [60]. Consumers generally tolerate coatings that are invisible and leave no flavor residues; however, any perceivable “film” could reduce purchase intent. Transparency in labeling and clear communication about the coating’s natural origin, eco-friendliness, and health safety can enhance consumer acceptance.

**Table 5 plants-15-00132-t005:** Impact of edible coatings on nutritional and sensory quality.

Coating System	Produce	Key Quality Attributes Preserved	Sensory & Nutritional Outcomes	Reference
Carnauba wax (lipid-based)	Moro oranges	Firmness, anthocyanins, vitamin C	Glossy appearance; slight antioxidant decline	[60]
Alginate + avocado pulp extract	Minimally processed apples	Browning inhibition, color, firmness	High sensory scores over 15 days; no waxy mouthfeel	[72]
Seaweed (*K. alvarezii*) coating	Mango	Weight loss reduction, texture	Neutral sensory impact; texture retained	[67]
Alginate oligosaccharide coating	Fresh cut papaya	Color retention, firmness, fungal suppression	Fresh appearance; no taste alteration	[40]
Aloe vera and Moringa oleifera plant extract edible coatings with chitosan nanoparticles	‘Cavendish’ bananas	TSS, fruit firmness, and peel color	highest score for color, texture, odor, and overall acceptability.	[42]
Pectin + lemongrass oil + CNC	Strawberries	Color, firmness, antioxidant retention	Slight surface sheen; no off flavors	[32]
Xanthan gum + Spirulina + pomegranate seed oil	Limes	Weight loss control, phenolics, PPO inhibition	Smooth texture; aroma unaffected	[46]
Alginate–based	Fresh cut pineapple	Shelf life	Color, texture and pH, were better preserved in the treated (coated)	[64]

WL = weight loss; PPO = polyphenol oxidase; CNC = Cellulose nanocrystals.

Overall, the functionality of edible coatings in fresh-fruit preservation extends beyond single-attribute barrier effects and should be understood as a multi-functional system operating at the interface between material properties and fruit physiology. As reviewed in this section, edible coatings contribute to quality preservation through (i) regulation of moisture loss via modification of water vapor transport, (ii) modulation of gas exchange and respiration dynamics to delay ripening and senescence, (iii) suppression of microbial growth through intrinsic bioactivity or incorporation of antimicrobial agents, and (iv) preservation of nutritional and sensory attributes by limiting oxidative degradation and enzymatic browning. Importantly, these functionalities are interdependent, and enhancement of one function may compromise another if coating composition and structure are not carefully optimized. The reviewed studies collectively indicate that effective edible coating design requires balancing transport properties, antimicrobial efficacy, and sensory acceptability in relation to fruit-specific physiology and storage conditions, rather than maximizing any single functional attribute in isolation.

## 4. Strategies for Optimizing Edible Coatings in Fresh Fruit Preservation

### 4.1. Material Selection and Formulation

Application strategies for edible coatings play a critical role in determining coating uniformity, adhesion, functional performance, and industrial feasibility. In fresh-fruit postharvest systems, application methods must accommodate high product throughput, variable fruit geometry and surface properties, and compatibility with existing packing-line operations. Consequently, the effectiveness of a coating formulation cannot be evaluated independently of its application strategy, as process-related factors such as residence time, shear forces, drying kinetics, and coating thickness strongly influence barrier properties and functional outcomes. This subsection focuses on application approaches most relevant to fresh-fruit handling, with emphasis on their practical advantages, limitations, and scalability.

Table 6 provides a structured synthesis of optimization strategies for edible coatings, linking material selection, formulation, application methods, and analytical evaluation with functional performance and industrial relevance. From a postharvest systems perspective, the practical adoption of edible coatings is constrained by factors such as packing-line speed, fruit-to-fruit variability in size and surface morphology, drying time limitations, and compatibility with existing cold-chain operations. While laboratory-scale studies often demonstrate promising functional performance, scaling these approaches requires careful consideration of coating uniformity, process control, and throughput efficiency. As summarized in Table 6, optimization strategies must therefore balance material functionality with application feasibility to ensure that coating technologies can be realistically integrated into commercial fresh-fruit handling systems. The functional performance of edible coatings, including barrier efficiency, antimicrobial activity, and sensory neutrality, is largely dictated by the type and combination of materials used. Edible coatings are constructed primarily from biopolymer materials such as polysaccharides (alginate, chitosan, pectin, cellulose derivatives), proteins (zein, gelatin), and lipid-based components (beeswax, carnauba wax). These materials form the structural matrix of coatings and films, enabling them to act as moisture and gas barriers while providing functional sites for antimicrobial or antioxidant incorporation. Their widespread use in fruit and vegetable preservation is well reflected in studies such as [27,28], which collectively highlight their versatility and compatibility with fresh-fruit surfaces.

To enhance functional performance, bioactive additives—including essential oils, metal nanoparticles such as ZnO or Ag, and plant extracts—are often incorporated into these biopolymer matrices. These compounds are valued for their antimicrobial and antioxidant properties, enabling extended shelf life and reduced microbial spoilage. Representative examples include silver-nanoparticle composites, Spirulina-enriched coatings, and essential-oil–reinforced chitosan films, as demonstrated in [46,58,62]. For example, chitosan coatings enriched with clove essential oil have been shown to significantly suppress microbial proliferation and delay quality deterioration in minimally processed strawberries, highlighting the synergistic antimicrobial action of chitosan and phenolic EO constituents such as eugenol [74]. The preparation of these coatings relies on robust formulation methods, including solution casting, homogenization of emulsions, nanoparticle dispersion, and ultrasonication. These processes are critical for achieving stable, homogeneous coating systems, ensuring proper distribution of active compounds within the matrix. Once formulated, edible coatings are commonly applied to produce via dipping, spraying, or brushing. These simple, scalable techniques enable uniform deposition across variable fruit surfaces and can be integrated into commercial processing operations. Their practical relevance is evident in recent experimental applications. Memete et al. [71] demonstrated that dipping-based application of polysaccharide coatings on fresh black mulberry fruit (*Morus nigra*) resulted in uniform surface coverage, significantly delaying softening and decay during cold storage. In a related context, Sarengaowa et al. [58] applied chitosan–cinnamon oil coatings to fresh-cut potatoes via controlled immersion, showing that coating uniformity was critical for achieving consistent antimicrobial efficacy and moisture control across cut surfaces. Similarly, Mohammadi et al. [46] employed standardized dipping and spraying protocols for xanthan gum–based composite coatings on Mexican lime, where controlled application ensured reproducible coating thickness and contributed to reduced weight loss, delayed enzymatic browning, and improved antioxidant retention. Collectively, these studies illustrate how application method and process control directly influence coating performance and reproducibility under practical postharvest conditions.

Evaluating coating performance often begins with moisture barrier analysis, typically through water vapor permeability (WVP) or WVTR measurements. These analyses quantify the coating’s ability to reduce moisture loss and thus prevent textural degradation in produce. The significance of WVP in edible film optimization is highlighted in studies such as [32,44], where nanocellulose- and chitosan-based films demonstrate measurable improvements in barrier properties. Gas exchange behavior is equally critical. Gas permeability analysis, including O_2_/CO_2_ transmission or headspace measurements, ensures that coatings do not impede normal respiration to the point of inducing anaerobiosis or off flavors. Reviews and experimental studies such as [16,72] provide methodological frameworks for understanding these gas-exchange dynamics within coated or MAP-treated fresh fruits.

Mechanical performance is assessed through tensile strength, elongation, and modulus testing, which determines a film’s durability, flexibility, and suitability for handling. Mechanical characterization is especially important for films that must withstand stacking, abrasion, or packaging stress. Such analyses are prominently discussed in [32,48], demonstrating how structural modifications influence mechanical integrity. The antimicrobial effectiveness of coatings is verified through microbiological assays such as zone-of-inhibition testing, microbial counts, or in situ challenge studies. These tests confirm the coating’s ability to suppress spoilage organisms or pathogens, as documented in [56,58], where nano-enhanced or essential-oil-enriched coatings effectively reduced microbial loads during storage. Coating evaluation also involves nutritional and chemical analyses, typically using spectrophotometry or HPLC to quantify vitamin C, phenolics, and antioxidant capacity. These methods help assess how coatings preserve nutritional quality over time. For example, Refs. [62,69] provide detailed insight into antioxidant retention and biochemical stability in coated apples and citrus.

Sensory evaluation—via consumer or trained panels—plays a crucial role in determining a coating’s acceptability. These assessments identify potential defects such as waxiness, off-flavors, or undesirable textures. Sensory methodologies and outcomes are thoroughly described in [68,69], which show how coatings influence appearance, aroma, flavor, and overall consumer preference. Finally, advanced analytical techniques such as SEM, FTIR, XRD, and DSC/TGA provide deep insight into microstructure, chemical interactions, crystallinity, and thermal stability. These tools support the development of improved formulations and structure–property relationships. Key examples include nanocellulose-reinforced and essential-oil composite films studied in [32,37,48], demonstrating how these techniques drive innovation in edible-coating materials science.

Overall, effective implementation of edible coatings in fresh-fruit supply chains depends not only on formulation design but also on selecting application strategies that balance coating performance, process efficiency, and commercial practicality.

### 4.2. Coating Application Techniques

The method used to apply edible coatings to fruit and vegetable surfaces significantly influences the coating’s uniformity, functional performance, and industrial applicability. These application techniques determine the coating’s barrier properties, adhesion, and consistency, which in turn affect its effectiveness in preserving fresh fruits throughout the supply chain. Common methods include dipping, spraying, brushing, and layer-by-layer (LbL) deposition, while emerging innovations such as electrostatic spraying and ultrasonic atomization show promise for scalable precision.

#### 4.2.1. Dipping

Dipping remains the most prevalent in laboratory settings due to its simplicity and full-surface coverage. It typically involves immersing the produce in a 1–3% (*w*/*v*) coating solution for 1–5 min, followed by drying at 20–30 °C for 30–60 min [46]. Dipping achieves coating thicknesses of 10–100 µm, influenced by solution viscosity and immersion time. The process is underpinned by wetting dynamics described by Young’s equation:(1)$$cos\theta =\frac{\gamma_{SV}-\gamma_{SL}}{\gamma_{LV}}$$

The interfacial interactions involved are illustrated in Figure 2, which shows a liquid droplet on a solid surface in equilibrium with the vapor phase, including the relevant surface tensions (γ_SV_, γ_SL_, γ_LV_) and contact angle (θ) as defined by Equation (1) θ, measured through the liquid, represents the angle at which the liquid-vapor interface meets the solid surface, indicating the degree of wetting where θ < 90° suggests good wetting (hydrophilic behavior), while θ > 90° indicates poor wetting (hydrophobic behavior). The interfacial tension between the solid and vapor phases, γ_SV_, reflects the surface energy of the solid in the absence of the liquid, while γ_SL_, the interfacial tension between the solid and liquid phases, depends on interactions such as hydrogen bonding or van der Waals forces between the coating solution and the fruit cuticle. Lastly, γ_LV_, the interfacial tension between the liquid and vapor phases, is equivalent to the surface tension of the coating solution, typically ranging from 30 to 70 mN/m for aqueous polysaccharide solutions.

#### 4.2.2. Spraying

Spraying is widely regarded as the most scalable edible-coating application method for fresh fruit because it enables continuous operation, reduced solution consumption, and controllable coat mass per unit area. In contrast to dipping, spray coating minimizes cross-contamination risk and avoids dilution/drag-out losses, while allowing coating thickness to be tuned via nozzle type, atomization pressure, flow rate, droplet size distribution, and spray time. Comparative evidence indicates that coating method measurably affects uniformity and final barrier function; for example, studies directly comparing dipping and spraying on fresh-cut fruit show method-dependent differences in nutritional retention and surface coverage, attributable to differences in deposited solids and drying dynamics [75]. Recent work comparing dip, brush, spray, and electrostatic spray further demonstrates that the application route can be as influential as formulation choice in determining coating quality and preservation outcomes [76]. These results support the need to treat “application method” as a first-order design factor alongside polymer class and additives.

#### 4.2.3. Electrostatic Spraying

Electrostatic spraying improves deposition efficiency by electrically charging droplets so they are attracted to the grounded produce surface, increasing wrap-around coverage and reducing overspray losses. This approach is particularly attractive for industrial lines where coating solution cost and wastewater handling are constraints, and where uniform deposition on complex surfaces is difficult with conventional sprays. Prototype/engineering studies for fruit coating systems report improved transfer efficiency and more complete surface coverage versus uncharged spray under comparable operating conditions [77]. In preservation-oriented comparisons, electrostatic spray performance has been shown to be comparable to (or better than) conventional spray for key quality outcomes, while using thinner coatings and shorter drying times [76]. Collectively, these studies position electrostatic spraying as a promising scale-up option, but they also highlight the need for commodity-specific optimization of charge-to-mass ratio, nozzle–fruit distance, line speed, and grounding strategy.

#### 4.2.4. Ultrasonic Coating

Ultrasonic coating uses high-frequency vibration to generate fine droplets with a narrow size distribution, enabling highly uniform thin films at relatively low flow rates and reduced clogging risk for certain biopolymer solutions. This is particularly relevant when nanocellulose, emulsions, or bioactive-loaded formulations are used, where shear history and droplet size can influence phase separation and final microstructure. A recent primary study designing and optimizing an ultrasonic coating system for fresh fruit demonstrated that process parameters (e.g., flow rate, power, exposure time) significantly affected coating uniformity and shelf-life outcomes, supporting ultrasonic atomization as a precision deposition route rather than merely an alternative spray head [78]. For high-value fruits or active coatings where dose control matters, ultrasonic deposition offers a practical pathway to reduce variability and improve reproducibility relative to conventional spray.

#### 4.2.5. Fluidized-Bed Coating

Fluidized-bed coating is used extensively in food/pharma for coating particulates because it provides intense convective drying and uniform exposure of moving particles to atomized coating droplets. While it is less suited to intact fresh fruit, it is directly relevant for small fruit products (e.g., raisins/dried berries) and coated inclusions, and it provides a useful process analog for continuous coating–drying lines. Primary postharvest-oriented studies on dried fruit demonstrate that edible coatings can be applied to raisins to improve texture/appearance and slow quality loss during storage, highlighting the practical relevance of controlled deposition and drying in particulate products [79]. In the context of fresh-fruit coatings, fluidized-bed concepts are valuable for understanding how air temperature, humidity, droplet flux, and residence time govern film formation, tackiness, agglomeration risk, and final barrier properties—factors that also constrain industrial spray lines for fresh produce.

#### 4.2.6. Panning Coating

Panning is a high-throughput deposition method in which products tumble in a rotating pan while coating solution is sprayed or metered, followed by forced-air drying. Although most commonly associated with confectionery and nuts, it is conceptually relevant for round/small produce items and for coated inclusions, because it couples mechanical mixing with staged application–drying cycles, enabling thicker multi-pass layers than single-pass spraying. In the edible-coating context, panning is therefore best viewed as a scale-up platform for batch coating of spherical items, where film build-up, tack control, and drying rate must be managed to avoid sticking and non-uniform thickness.

#### 4.2.7. Three-Dimensional Printing and Patterned Deposition

Three-dimensional printing is not yet a mainstream postharvest coating method for intact fruit; however, it is increasingly explored for structured edible layers and customized edible “skins” in fruit-based products. Primary studies show that fruit-based matrices can be printed into stable edible structures with measurable retention of bioactives/antioxidant properties under controlled printing conditions [80]. From an edible-coating perspective, the relevance of 3-D printing lies in precision patterning (localized barrier placement), multi-material deposition, and the ability to engineer thickness gradients—capabilities that could inform future high-value coating applications where conventional dipping/spraying cannot easily control spatial functionality.

#### 4.2.8. Brushing

Brushing is employed experimentally or for irregular surfaces, applying coatings using manual or mechanical brushes, typically producing coating thicknesses in the order of 20–60 µm, with surface-dependent variability that can exceed 20% depending on formulation and handling [75,76,77]. The method relies on frictional force and manual precision, lacking scalability for industrial use. Its theoretical foundation is based on contact mechanics, where coating distribution depends on brush pressure, bristle texture, and solution rheology, particularly viscosity and shear-thinning properties [49]. Due to its dependence on operator technique, this method can lead to inconsistency, especially for produce with complex geometries like strawberries or mulberries [34]. Brushing is generally reserved for small-scale applications or laboratory settings where other techniques are not feasible.

Although simple and low-cost, the technique is less efficient, showing lower repeatability and throughput compared to dipping or spraying. Bharti et al. [49] observed greater variability in coating thickness and phenolic compound retention for brushed samples of mangoes compared to sprayed samples. These findings underline that while brushing may serve niche applications, it is not ideal for large-scale operations where uniformity and throughput are critical.

Furthermore, surface energy of the fruit, coupled with contact angle behavior, influences coating spread and adhesion during brushing. As with other methods, brushing is sensitive to environmental factors such as temperature and humidity, which alter coating viscosity and drying rate. Brushing is thus best viewed as a supplementary or pilot-stage method, useful for experimental formulations but suboptimal for postharvest logistics on a commercial scale.

#### 4.2.9. Layer-by-Layer (LbL)

LbL deposition involves sequential application of oppositely charged biopolymer layers (e.g., cationic chitosan and anionic alginate) via dipping or spraying, enabling controlled multilayer architectures at the micro- to submicron scale [71]. The technique exploits electrostatic interactions and hydrogen bonding between oppositely charged layers, allowing tailored functionality and enhanced barrier performance. Memete et al. [71] showed that sequential application of lipid- and gelatin-based layers (LbL coating) on mulberries significantly enhanced phenolic stability and slowed physicochemical quality degradation during refrigerated storage relative to single-layer treatments.

LbL systems offer advantages in controlled release of bioactives and multilayered protection against moisture and gas exchange. Bharti et al. [49] also highlighted the potential of multilayer systems in starch-based matrices for improving antimicrobial efficacy, showing over 90% inhibition against Gram-positive bacteria after during storage. However, the process is inherently time-intensive and requires precise control of solution chemistry and processing conditions, which limits throughput and increases operational complexity relative to single-step coating methods. The LbL process benefits from optimization of interlayer adhesion, pH control, and ionic strength of the coating solutions to avoid delamination. Temperature and humidity also affect layer stability during storage, particularly in coatings incorporating thermosensitive ingredients. Despite its labor-intensiveness, LbL is highly promising for research and premium applications where multifunctionality and precision are prioritized over scale.

### 4.3. Analytical and Evaluation Methods

Thorough evaluation of edible coatings involves a range of physical, chemical, microbiological, and sensory tests. These assessments are necessary to determine barrier properties, nutritional retention, microbial inhibition, structural integrity, and consumer acceptability.

#### 4.3.1. Barrier Properties

The water vapor transmission rate (WVTR) is a fundamental metric for moisture control in edible coatings and packaging films. It is commonly measured using gravimetric cup methods, such as the Desiccant and Water Methods, under standardized conditions defined by ASTM E96/E96M-23. These tests quantify vapor flux by monitoring weight changes in coated samples under controlled temperature and humidity. In a study by Pizato et al. [74], strawberries coated with 2% chitosan enriched with 1.5% clove essential oil exhibited significantly reduced weight loss, 13.51% compared to 24.19% in uncoated controls, over 12 days of refrigerated storage. This reduction in moisture loss was attributed to the coating’s semi-permeable barrier properties, which effectively modulated WVTR while preserving texture and color. Gas permeability, particularly for oxygen (O_2_) and carbon dioxide (CO_2_), is equally critical for assessing the impact of coatings on fruit respiration and anaerobic risks. As reviewed by Sánchez-Tamayo et al. [73], gas permeability in edible films is typically measured using manometric, gravimetric, or continuous-flow techniques, many of which are adapted from ASTM standards. These methods involve placing the film between two compartments, one exposed to the test gas and the other connected to a detector and quantifying the transmission rate under controlled conditions. The review emphasizes that permeability results are highly sensitive to film preconditioning, test setup, and environmental parameters, underscoring the need for standardized protocols when evaluating barrier performance in postharvest applications.

#### 4.3.2. Mechanical Behavior

Mechanical behavior is an essential but often underinterpreted dimension of edible-coating performance. Although coatings applied on fruit surfaces typically form thin layers in the 5–20 µm range, mechanical characterization is almost universally conducted on free-standing cast films—usually 50–200 µm thick—which serve as model systems for assessing the intrinsic strength and flexibility of coating formulations. While these films differ in thickness from applied coatings, their tensile properties provide meaningful insight into how biopolymer networks respond to deformation, bending, and surface stress encountered during handling, transport, and storage.

Across literature, tensile strength (TS) and elongation at break (EAB) remain the most consistently reported parameters. Table 7 summarizes the mechanical properties that are currently available from the studies reviewed here, highlighting both the strengths of existing characterization efforts and the persistent gaps—particularly the near-total absence of stiffness data—within edible-coating research. Starch–carrageenan composite films, for example, display moderate tensile strength (≈15 MPa) and relatively high elongation (≈28%), reflecting a balanced, ductile structure suitable for flexible coatings [48]. Chitosan-based systems reinforced with bacterial nanocellulose show substantially higher tensile strength, reaching more than 40 MPa at optimal filler loading—accompanied by reduced elongation, indicating a transition toward greater rigidity and reduced ductility [32]. Cellulose–chitosan blends exhibit tensile strengths in the 12–14 MPa range, illustrating the diversity of mechanical responses achievable through biopolymer blending [30]. These patterns emphasize how formulation strategies, such as introducing nanofillers or combining polysaccharides, directly shape the mechanical robustness and flexibility of edible films.

In contrast to TS and EAB, Young’s modulus—central to quantifying stiffness and deformation resistance—remains largely absent from edible-coating research. Among the studies surveyed, none reported absolute modulus values, and only a single review summarized modulus changes in relative terms (e.g., an ≈87% increase with cellulose nanocrystal addition) without providing corresponding baseline values [45]. This omission limits the development of structure–property models and hinders comparison across formulations, even though stiffness plays a crucial role in determining whether coatings crack, resist bending, or maintain integrity on curved, expanding, or mechanically stressed produce surfaces. As edible coatings evolve toward more engineered, multifunctional systems, routine reporting of Young’s modules would significantly enhance analytical and optimization capabilities.

Most mechanical tests follow ASTM D882, which standardizes specimen dimensions and tensile loading procedures for thin films, enabling reproducible reporting of TS and EAB. Nevertheless, relying solely on these two parameters underrepresents the mechanical complexity of edible coatings and constrains our ability to predict real-world performance. Integrating modulus measurement and, where possible, complementary techniques such as nanoindentation or flexural analysis would provide a more complete mechanical profile of coating materials. Figure 3 illustrates the mechanical trade-offs observed in chitosan-, CNC-reinforced-, and alginate-based edible coatings as plasticizer content increases. All systems show a consistent pattern: tensile strength declines while elongation at break rises. CNC-reinforced films retain higher tensile strength at comparable plasticizer levels, highlighting the structural benefits of nanocrystal incorporation. These trends, drawn from studies using ASTM D882 protocols [32,40,46], underscore the need to balance rigidity and flexibility when designing coatings for produce with varying mechanical sensitivities.

The durability of produce coatings is influenced by their composition, including the type of biopolymer, plasticizers, and additives used. For instance, Tabassum & Khan [40] demonstrated that incorporating glycerol as a plasticizer in alginate-based coatings increased elongation at break by 30% compared to non-plasticized films, though it slightly reduced tensile strength. This trade-off is critical, as overly rigid coatings may crack under stress, while excessively flexible coatings may fail to provide adequate protection. Coatings with high tensile strength and moderate elongation at break are ideal for produce like apples or tomatoes, which are prone to mechanical damage during bulk handling. Conversely, softer fruits like berries may require coatings with higher flexibility to accommodate surface deformation. Additionally, environmental factors such as humidity and temperature during storage and transport can affect coating performance. Liyanapathiranage et al. [35]) found that edible coatings maintained higher Young’s modulus values (up to 1.2 GPa) under low-humidity conditions, ensuring better resistance to deformation during long-distance shipping.

#### 4.3.3. Microbiological Efficacy

Microbiological efficacy is a critical factor in evaluating the performance of edible coatings enriched with antimicrobial agents, especially for extending the shelf life and ensuring the safety of perishable produce. Zone-of-inhibition assays are widely used to measure the antimicrobial activity of coatings by observing the clear zones around coated samples where microbial growth is inhibited. Bharti et al. [49] employed the disk diffusion method (Microan+51) to assess the efficacy of caraway EO-incorporated starch films, reporting significant inhibition zones against *B. cereus* and *S. aureus* with zones increasing with higher EO concentrations (e.g., up to 16 mm for *B. cereus* at 3% EO). Total plate counts quantify viable microbial loads, offering a direct measure of reduction over time, while challenge studies simulate real-world contamination, providing robust efficacy data. Bizymis et al. [62] achieved a 99% reduction in *E. coli* within 24 h using silver nanoparticles, complementing Bharti et al.’s findings on Gram-positive bacteria sensitivity. The antimicrobial performance of coatings depends on the type, concentration, and compatibility of the antimicrobial agent with the coating matrix. Bharti et al. [49] found that caraway EO, rich in cicerain (55.74%) and carvone (8.36%), exhibited greater efficacy against gram-positive bacteria (*B. cereus* and *S. aureus*) due to their thinner cell walls, which are more susceptible to phytochemicals, compared to the intrinsic tolerance of Gram-negative bacteria (*E. coli* and *P. aeruginosa*). This aligns with Bharti et al. [49] who tested levels of 0.5%, 1%, 2%, and 3% (TC1 to TC4), with higher concentrations showing enhanced inhibition (*p* < 0.01). Environmental factors like temperature and humidity further influence efficacy. For instance, Congying et al. [48] observed reduced performance of ZnO nanoparticle coatings at higher temperatures Bharti et al. [49] reported zone-of-inhibition assay results demonstrating clear antimicrobial activity of caraway essential oil (EO)-incorporated starch-based films against Bacillus cereus, Escherichia coli, Pseudomonas aeruginosa, and Staphylococcus aureus. The study highlighted that gram-positive bacteria exhibited larger inhibition zones as EO concentration increased. For comparison, Bizymis et al. [62] also presented similar findings when evaluating the efficacy of silver nanoparticles.

#### 4.3.4. Nutritional and Biochemical Analysis

Nutritional and biochemical analysis techniques are critical for assessing the preservative efficacy of edible coatings applied to perishable produce. These methods quantify antioxidant capacity, vitamin and phenolic content, and visual quality attributes, furnishing insight into how coatings mitigate oxidative degradation and nutrient loss during storage. Widely used assays include DPPH and FRAP for antioxidant evaluation, HPLC for micronutrient profiling, and colorimetric methods for monitoring browning and pigment degradation. The DPPH (2,2-diphenyl-1-picrylhydrazyl) and FRAP (Ferric Reducing Antioxidant Power) assays are prominent techniques used to measure the antioxidant potential of edible coatings. DPPH evaluates the ability of a sample to scavenge free radicals by observing the decrease in absorbance at 517 nm, while FRAP assesses reducing power through colorimetric change at 593 nm. For example, Bharti et al. [49] used the DPPH assay to demonstrate that starch-based films enriched with caraway essential oil-maintained antioxidant activity in coated fruit.

High-Performance Liquid Chromatography (HPLC) provides precise quantification of sensitive nutrients such as vitamin C and phenolic compounds. HPLC analyses often use a C18 column and UV detection to identify and quantify key bioactive molecules in fruit samples. Mohammadi et al. [46] used HPLC to show how chitosan coatings helped retain vitamin C content under cold storage conditions.

Colorimetric tests are also widely applied to monitor enzymatic browning and pigment degradation. Absorbance at 420 nm is typically used for browning index, while chlorophyll and anthocyanin retention are evaluated via spectrophotometric measurements at wavelengths such as 645 nm and 663 nm. Sun et al. [46] employed such techniques to examine pigment retention in tomato samples coated with chitosan–thyme oil films.

Analytical results are influenced not only by the coating composition but also by sample preparation, solvent selection, and storage conditions. Bharti et al. [49] extracted phenolic compounds using methanol and noted how higher concentrations of essential oil led to greater inhibition of browning. The barrier properties of the coating, such as thickness and oxygen permeability, also affect the preservation of labile nutrients. Mohammadi et al. [46] highlighted improved vitamin retention in thicker chitosan layers. Environmental parameters during storage further impact outcomes. Congying et al. [48] demonstrated that ZnO-enhanced starch coatings preserved anthocyanins better at 4 °C compared to ambient conditions, emphasizing the need to pair biochemical tests with storage simulations.

Standardized protocols enhance reproducibility and facilitate cross-study comparisons. AOAC and ISO guidelines often inform method selection. For instance, absorbance-based readings for browning or pigment retention follow standardized wavelength references, and chromatographic methods adhere to validated column and mobile phase parameters. Bharti et al. [49], Mohammadi et al. [46], and Sun et al. [50] all followed such protocols, ensuring scientific rigor.

#### 4.3.5. Sensory Evaluation

Sensory evaluation plays a pivotal role in determining consumer acceptance of coated fresh fruits, focusing on attributes such as appearance, taste, aroma, texture, and overall preference. This evaluation relies on structured methodologies to provide objective insights into how coatings influence sensory quality. Trained sensory panels and consumer groups are commonly employed, utilizing tools like hedonic scales and descriptive analysis to systematically assess these attributes. The process involves controlled settings to ensure consistency, with methodologies often aligned with international standards such as those from the International Organization for Standardization (ISO).

Hedonic scales, typically ranging from 1 (dislike extremely) to 9 (like extremely), allow panelists to rate overall liking and individual attributes like taste or appearance. Descriptive analysis, a more detailed technique, involves trained panelists who identify and quantify specific sensory characteristics, such as firmness, aroma intensity, or visual clarity, using standardized lexicons. Memete et al. [71] utilized both approaches, conducting evaluations over 8 days of refrigerated storage to track changes in sensory properties of coated black mulberries.

The choice of sensory evaluation method depends on the study’s objectives and the coating’s properties. Hedonic scales are ideal for consumer acceptance studies, capturing broad preferences, while descriptive analysis suits detailed profiling of sensory changes over time or across formulations. Environmental factors, such as storage conditions (e.g., refrigerated at 4 °C in Memete et al. [71], and the coating’s thickness or composition influence the sensory attributes assessed, necessitating adjustments in methodology. Sensory evaluation should adhere to standardized guidelines, such as ISO 8589 [81] for panel selection and training, and ISO 4121 [82] for sensory analysis methods. These standards dictate the design of evaluation sessions, including sample preparation (e.g., cutting uniform pieces of coated produce) and presentation order to avoid carryover effects. Memete et al. [71] likely followed similar protocols, ensuring that assessments of gelatin, oil, and wax coatings were conducted consistently across the 8-day period. Data analysis often incorporates software like FIZZ (Biosystèmes, Dijon, France) or Compusense (Compusense Inc., Guelph, ON, Canada) to manage scores and perform statistical tests, ensuring compliance with rigorous scientific practices.

#### 4.3.6. Advanced Structural Analysis

Techniques like scanning electron microscopy (SEM) provide insight into surface uniformity and porosity; FTIR spectroscopy identifies chemical bonding and interactions; and differential scanning calorimetry (DSC) assesses thermal stability. These methods enhance understanding of structure–function relationships and support the design of more stable and functional coating systems [46]. Despite the broad toolkit available, a key research gap remains in standardizing analytical protocols across studies to enable direct comparison of results. Moreover, real-world validation of coating performance under commercial cold chain logistics is still insufficient, and more studies are needed to evaluate behavior under fluctuating temperature and humidity conditions.

Scanning electron microscopy (SEM) is employed to examine the surface morphology, uniformity, and porosity of coatings at a microscopic level. Samples are typically prepared by freeze-drying or gold sputtering to enhance conductivity, followed by imaging under high vacuum at magnifications ranging from 100× to 10,000×. FTIR spectroscopy analyzes chemical bonding and intermolecular interactions by measuring the absorption of infrared light, with spectra recorded over a range of 400–4000 cm^−1^ using attenuated total reflectance (ATR) mode. Differential scanning calorimetry (DSC) evaluates thermal stability and phase transitions by heating samples (e.g., 0–200 °C) at a controlled rate (e.g., 10 °C/min), detecting endothermic or exothermic changes. Mohammadi et al. [46] utilized these techniques to characterize coating structures, providing a foundation for optimizing design parameters.

The choice of structural analysis method depends on the coating’s composition and the property of interest. SEM is ideal for visualizing physical structure, requiring careful sample preparation to avoid artifacts. FTIR suits the study of chemical compatibility, particularly for coatings with polysaccharides or lipids, while DSC is essential for thermal-sensitive materials like protein-based films. Sample size, moisture content, and instrument calibration influence results, necessitating standardized preparation protocols. Environmental conditions during analysis, such as temperature and humidity, also affect outcomes, highlighting the need for controlled settings. Structural analysis follows guidelines from organizations like the International Union of Pure and Applied Chemistry (IUPAC) and ASTM International. SEM protocols specify sample mounting and vacuum levels, while FTIR adheres to standards for spectral resolution (e.g., 4 cm^−1^) and scan numbers (e.g., 32 scans). DSC testing aligns with ISO 11357, detailing heating rates and sample pans. Mohammadi et al. [46] likely adhered to these standards, though variations in protocols across studies complicate comparisons. Efforts to harmonize methodologies, such as adopting universal sample preparation or data reporting formats, are underway but not yet fully realized.

Taken together, the practical implementation of edible coatings in fresh-fruit systems is constrained by a set of interdependent material, mechanical, and process-related trade-offs. Enhancing coating strength or barrier performance through polymer reinforcement or filler incorporation often leads to increased stiffness and reduced extensibility, which can elevate the risk of cracking, delamination, or failure under handling and transport stresses. Conversely, highly flexible coatings may lack sufficient mechanical integrity to withstand abrasion, compression, or repeated contact during postharvest operations. As illustrated in Table 8, reported tensile strength, elongation behavior, and Young’s modulus (or stiffness trends where absolute values are unavailable) reveal non-linear responses to formulation changes, indicating narrow optimization windows rather than monotonic performance gains. These mechanical constraints are further compounded by application-related stresses, variability in fruit surface morphology, and scalability requirements within commercial packing lines. Collectively, these factors underscore the need to evaluate edible coatings as integrated material–process systems, where formulation design, application method, and mechanical robustness must be optimized concurrently to achieve reliable performance under real postharvest conditions.

## 5. Environmental Concerns and Limitations

### 5.1. Environmental Footprint and End-of-Life Considerations

Edible coatings have emerged as promising technology to extend the shelf life and maintain the quality of fresh fruits within the supply chain. However, their integration into large-scale agricultural and commercial systems raises significant environmental concerns and operational limitations. Table 6 shows the environmental and operational metrics of edible coatings compared to synthetic plastics. These challenges include the ecological footprint of raw materials, the end-of-life disposal of coated produce, and the scalability of production processes.

The production of edible coatings involves raw materials such as polysaccharides (e.g., chitosan, alginate), proteins, and lipids, which are derived from natural or agricultural sources. Although biodegradable, the cultivation and processing of these materials can contribute to environmental degradation. For instance, Mohammadi et al. [46] noted that chitosan production entails deacetylation of chitin from crustacean shells, which requires considerable chemical and thermal inputs, increasing emissions. Similarly, large-scale extraction of alginate from brown seaweed, if not managed sustainably, may impact marine ecosystems [48]. Starch-based coatings made from crops like corn or cassava are water-intensive and may compete with food production, raising sustainability concerns [49].

The disposal of coated produce also presents challenges. Although edible coatings are biodegradable under composting conditions, field degradation is often incomplete. For example, Bharti et al. [49] reported partial breakdown of polysaccharide-based coatings after 90 days under controlled composting, with incomplete mineralization in high-humidity landfill scenarios. This may lead to temporary organic residue accumulation, although still significantly lower than the persistence of synthetic packaging materials. In contrast, polyethylene liners used in fresh fruits packaging are durable but contribute to long-term plastic pollution.

### 5.2. Operational Scalability and Process Limitations

Scalability and process efficiency remain major constraints in the commercial deployment of edible coatings. Industrial application methods—typically dipping, spraying, or brushing—must achieve consistent coating thicknesses (10–80 µm) across heterogeneous fruit surfaces [83]. However, equipment limitations, surface topography, and high line speeds often reduce uniformity at scale. Mohammadi et al. [46] reported a 20–30% decline in coating efficiency during high-throughput operations, largely driven by incomplete coverage on irregular or highly textured produce.

Despite these operational challenges, edible coatings are gaining momentum in postharvest systems. Synthetic wax coatings continue to dominate due to their long-established industrial use, compatibility with existing equipment, and predictable performance. However, recent advances in bio-based edible coatings—supported by growing evidence of their functional, physiological, and quality-preserving benefits [25]—are increasingly positioning them as strong alternatives for high-value commodities. These formulations offer improved barrier properties, opportunities for functionalization (e.g., antimicrobials, antioxidants), and enhanced alignment with sustainability-driven supply chain goals. Synthetic polymer coatings incur a notable carbon footprint during production, with cradle-to-grave greenhouse-gas emissions for polyethylene typically reported in the range of 2–3 kg CO_2_-eq kg^−1^, depending on grade and energy mix [19,84,85]. In contrast, edible coatings derive from renewable biopolymers but often require higher processing energy per unit mass due to extraction, purification, and formulation steps [48]. Thus, while edible coatings present a promising eco-aligned pathway, their sustainability benefits are closely tied to upstream processing efficiencies and material sourcing.

### 5.3. Regulatory Frameworks and Approval Constraints (U.S. and EU)

Regulatory and safety considerations significantly influence the adoption of edible coatings. Such coatings must comply with food safety regulations, including U.S. FDA requirements for direct food additives and Generally Recognized as Safe (GRAS) substances, which may impose substance-specific limitations on the use of active ingredients such as essential oils to avoid adverse sensory impacts. Bharti et al. [49] illustrated this trade-off by showing that reducing essential-oil concentrations to improve sensory acceptability diminished antimicrobial effectiveness in starch-based composite coatings.

Environmental performance is commonly evaluated using standardized life-cycle and biodegradability assessment frameworks, including ISO 14040/14044 for life-cycle assessment (LCA) and ISO 14855 for compostability testing [86,87]. Comparative studies applying these approaches indicate that chitosan- and cellulose-based coatings exhibit substantially higher biodegradability under composting conditions than polyethylene-based packaging, which shows negligible degradation over comparable timeframes [48,84]. Cradle-to-grave assessments further indicate that conventional polyethylene coatings are associated with greenhouse-gas emissions on the order of 2–3 kg CO_2_-eq per kilogram of polymer, depending on grade and energy mix [19,84,85].

The overall ecological footprint of edible coatings is therefore strongly dependent on raw-material sourcing, formulation complexity, and end-of-life handling. Approaches such as the valorization of agro-industrial by-products for biopolymer extraction and the use of sustainably managed biomass sources have been shown to reduce processing energy demand and improve environmental performance relative to petroleum-based materials [48]. Nevertheless, edible coatings still face challenges related to application efficiency, durability, and consistency at scale, while synthetic coatings remain constrained by end-of-life impacts and increasing regulatory and societal pressure to reduce plastic waste [19,84,85].

Beyond material-level considerations, regulatory frameworks further shape commercialization pathways. In the United States, edible coating components are regulated primarily as food additives or GRAS substances under FDA oversight, with additional scrutiny required for composite or active systems incorporating antimicrobials, antioxidants, or nanoparticles. In the European Union, regulatory oversight is more fragmented, as edible coatings may fall under food additive, food-contact material, or active-packaging regulations, depending on composition and intended function. This lack of harmonized approval pathways for multifunctional edible coatings across jurisdictions continues to represent a barrier to large-scale adoption, underscoring the need for regulatory-aware formulation design and standardized performance evaluation protocols.

## 6. Conclusions

Edible coatings, especially those derived from natural biopolymers, present a promising advancement in sustainable postharvest packaging for fresh fruits. Their functional versatility, including moisture and gas barrier properties, microbial inhibition, and sensory quality preservation, positions them as viable alternatives to synthetic packaging materials in fresh fruit supply chains. However, their effective implementation demands a detailed understanding of material properties, formulation strategies, application methods, and performance evaluation protocols.

This review has highlighted significant progress in the development and application of polysaccharide-, protein-, and lipid-based coatings, yet notable gaps remain in aligning coating performance with the physiological and logistical requirements of commercial postharvest systems. Key factors such as mechanical integrity, water vapor permeability, and gas transmission characteristics of biopolymer films directly impact a coating’s ability to function as viable packaging substitutes. Standard testing methods, such as ASTM D882 (for mechanical properties) and ASTM E96/E96M (for WVP), remain crucial for characterizing these materials. However, performance is often limited by the plasticizing effect of moisture in hydrophilic matrices, which reduces barrier integrity. This challenge underscores the need for targeted research into barrier enhancement techniques, such as nanostructuring, hydrophobic modification, or multilayer designs.

Importantly, edible coatings not only serve as passive barriers but can actively modulate the fruit’s internal atmosphere, potentially replicating or enhancing the microenvironmental benefits of modified atmosphere packaging. Yet, the interplay between coating properties and fruit physiology remains insufficiently explored, particularly under commercial handling conditions. Future studies must bridge this gap by integrating advanced material characterization with physiological monitoring and shelf-life assessments under real-world scenarios. Future research should therefore focus on integrating quantitative transport and mechanical characterization with fruit physiological responses under commercial postharvest conditions. In addition, greater emphasis is needed on scalable application technologies and standardized evaluation protocols, alongside lifecycle and regulatory considerations, to support industry adoption.

In conclusion, optimizing edible coatings for fresh fruits preservation requires a holistic approach that balances material innovation, application scalability, and postharvest efficacy. Advancing this field will depend on interdisciplinary collaboration across food science, material engineering, and postharvest technology, supported by standardized evaluation protocols and regulatory clarity. With such efforts, edible coating could transition from niche innovation to mainstream application in sustainable fresh fruits logistics.

## Figures and Tables

**Figure 1 plants-15-00132-f001:**
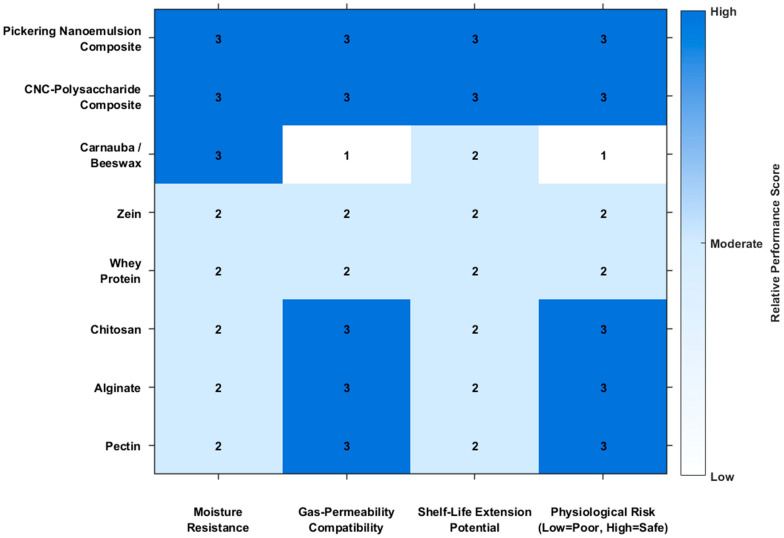
Heatmap comparing the performance of selected edible coating materials across three key attributes: moisture resistance, gas permeability, and shelf-life extension. Performance scores (1 = low, 2 = moderate, 3 = high) were derived from recent empirical studies and reviews. The scoring reflects moisture barrier efficiency (e.g., water vapor transmission reduction), gas exchange compatibility for respiration, and overall preservation effects on fruit quality.

**Figure 2 plants-15-00132-f002:**
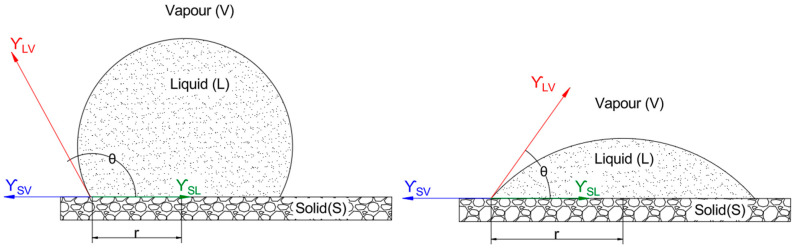
Illustration of a liquid droplet on a solid surface in equilibrium with the vapor phase, showing the interfacial tensions (γ_SV_, γ_SL_, γ_LV_) and the contact angle (θ) as defined by Young’s equation (Equation (1)). When the two materials come in contact, γSL represents the surface tension between them, forming the contact angle θ. A contact angle of less than 90° (**right**) indicates that wetting of the surface is favorable. Otherwise, A contact angle of greater than 90° (**left**), the wettability is unfavorable.

**Figure 3 plants-15-00132-f003:**
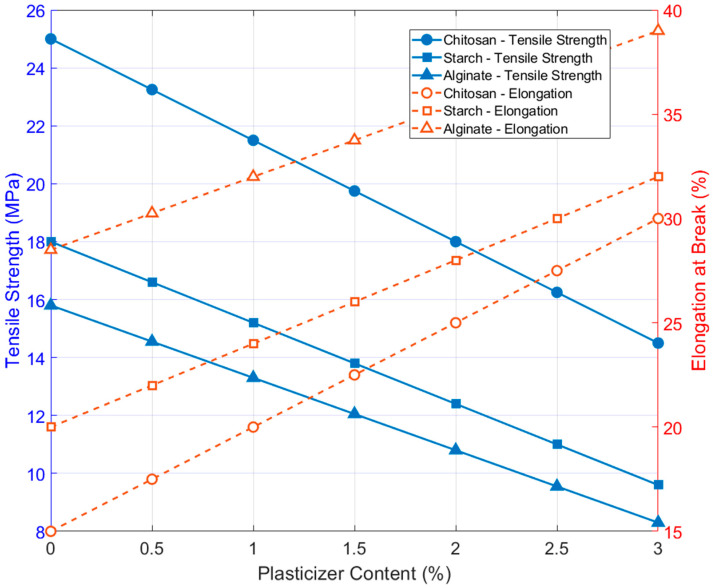
Relationship between plasticizer content and mechanical properties (tensile strength in MPa and elongation at break in %) of edible coatings composed of chitosan-, CNC-reinforced-, and alginate-based materials. Data trends are based on Mohammadi et al. [46] for chitosan-based coatings, Ref. [32] for nanocrystal-enhanced biopolymer systems, and [40] for alginate-based coatings, with all studies employing ASTM D882 protocols.

**Table 6 plants-15-00132-t006:** Strategies and evaluation frameworks for optimizing edible coatings in fresh-fruit preservation.

Optimization Domain	Materials/Techniques	Functional or Process Objective	Industrial or Practical Relevance	Reference
Biopolymer Material Selection	Alginate, chitosan, pectin, cellulose derivatives, zein, gelatin, beeswax, carnauba wax	Structural film formation; moisture and gas barrier control; carrier matrix for actives	Determines baseline coating performance and fruit compatibility	[46,59,60]
Bioactive Incorporation	Essential oils; nanoparticles (ZnO, Ag); plant extracts	Antimicrobial and antioxidant enhancement; shelf-life extension	Enables multifunctionality but requires controlled release and regulatory compliance	[46,58,62]
Formulation Engineering	Emulsion homogenization; nanoparticle dispersion; ultrasonication; plasticizer optimization	Stable coating structure; uniform distribution of active agents	Critical for reproducibility and scale-up consistency	[62,68]
Application Strategies	Dipping; controlled immersion	Uniform coating deposition; throughput compatibility	Must integrate with packing-line speed and fruit geometry	[46,58,71]
Moisture Barrier Optimization	WVP/WVTR measurement; thickness control	Minimize transpiration-driven weight loss	Directly linked to visual quality and marketability	[32,47]
Gas-Exchange Regulation	O_2_/CO_2_ permeability; headspace gas monitoring	Maintain aerobic respiration; delay ripening	Prevents anaerobiosis and off-flavor development	[16,73]
Mechanical Integrity	Tensile strength; elongation; modulus (ASTM D882)	Resistance to cracking, handling damage	Influences coating durability during transport	[32,49]
Antimicrobial Validation	Zone-of-inhibition assays; microbial counts; in situ challenge tests	Verify efficacy against spoilage organisms and pathogens	Supports safety claims and regulatory approval	[58,62]
Nutritional & Chemical Stability	HPLC; spectrophotometry; DPPH/FRAP assays	Retention of vitamins, phenolics, antioxidant capacity	Maintains nutritional value and consumer perception	[60,72]
Sensory Acceptance	Trained/consumer panels; hedonic scoring; descriptive analysis	Detect off-flavors, texture or appearance changes	Critical for market adoption	[71,72]
Structure–Property Characterization	SEM; FTIR; XRD; DSC/TGA	Link microstructure to transport and mechanical behaviors	Supports rational coating design	[32,37,49]

**Table 7 plants-15-00132-t007:** Mechanical properties of edible coatings for produce.

Material System	Tensile Strength (MPa)	Elongation at Break (%)	Young’s Modulus/Stiffness Response	Functional Implication for Coatings	Reference
Starch–carrageenan composite film (control)	15.23 ± 0.90	27.84 ± 2.59	Not reported	Moderate flexibility; limited resistance to handling stress	[49]
Pristine chitosan film	21.07 ± 1.64	33.84 ± 2.51	Not reported	Balanced strength–flexibility suitable for thin coatings	[32]
Chitosan + 2% BNC	27.03 ± 1.46	29.71 ± 2.15	↑ stiffness (trend reported)	Improved strength with acceptable flexibility	[32]
Chitosan + 4% BNC	41.32 ± 2.20	23.76 ± 1.52	High stiffness (optimum region)	High strength with increased brittleness risk	[32]
Chitosan + 6% BNC	34.75 ± 1.02	25.11 ± 2.93	↓ stiffness (aggregation effects)	Over-reinforcement reduces mechanical efficiency	[32]
Cellulose–chitosan (CeCh) film	12.40 ± 0.02	Qualitative (no significant change)	Not reported	Insufficient strength for standalone barrier use	[30]
Cellulose–chitosan + anthocyanin	14.00 ± 0.05	Not reported	Slight ↑ stiffness	Added functionality at cost of mechanical robustness	[30]
Chitosan + CNC (summary)	Formulation-dependent	Formulation-dependent	↑ modulus (~87% at 5% CNC); ↓ at higher loadings	Non-linear reinforcement defines narrow design window	[32]

↑ and ↓ denote increasing and decreasing trends, respectively, relative to the control or adjacent formulations, as reported in the referenced studies.

**Table 8 plants-15-00132-t008:** Environmental and operational metrics of edible coatings vs. synthetic plastics.

Metric	Edible Coatings	Synthetic Plastics	Reference
Carbon Footprint	Renewable biopolymers with lower long-term environmental burden.	Fossil-based polymers with higher production emissions.	[27,29]
Water Use	Water-intensive biopolymer extraction; varies by source.	Lower process water use but higher upstream fossil-fuel impact.	[18,27]
Degradation/End-of-Life	Biodegradable and compostable.	Non-biodegradable; persistent in environment.	[34]
Material Cost Characteristics	Higher processing input; often used for premium produce.	Widely available, lower cost, and industry-standard.	[20,25]
Application Efficiency	Coverage varies; sensitive to fruit surface variability.	Consistent mechanical protection with minimal variability.	[32,46]

## Data Availability

This review is based on previously published studies and does not include new experimental data. All sources referenced are publicly available and cited accordingly. Further information can be obtained from the corresponding author upon reasonable request.

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
