# Peer review of "Edible Coatings for Fresh Fruits: Functional Roles, Optimization Strategies, and Analytical Perspectives"

_plants, 2026, doi:10.3390/plants15010132_

Round 1
Reviewer 1 Report
Comments and Suggestions for Authors
This paper reviewed the functional roles, optimization strategies, and analytical perspectives of edible coatings in fresh agricultural products preservation. It offers important insights for the application of edible coatings in food preservation and packaging. However, the study exhibits relatively limited innovation, and its writing could be further refined. Specific comments are provided below:
1.This paper reviewed many aspects of edible coatings. However, personally, I feel that many parts have not been covered comprehensively and the depth is insufficient.
2.The term “fresh produce” in the title could be replaced with “fresh fruits” to better align with the review content, which focuses predominantly on fruits.
3.Abstract: It is suggested to appropriately add some textual content related to the key findings and conclusion.
4.The section entitled “Edible coating materials” lacks thoroughness and clarity. It does not provide a comprehensive review or summary of the major materials used in edible coatings (eg. polysaccharides, protein, lipids, composite etc.), nor does it adequately cover the minor functional additives (eg., surfactants, plasticizers, antioxidants or antimicrobial agents etc.) that can be incorporated.
5.Figure 1: The selection of the horizontal and vertical axes in my opinion is insufficient. Meanwhile, the corresponding text in the main manuscript needs to be revised.
6.Table 1-5 could be improved by expanding the list of materials and composite or coating system, as the current presentation appears somewhat limited in scope.
7.The functionality of the edible coating in the third part is not fully covered in the summary. The author is advised to provide more details.
8.Part 3: How do these edible coatings provide these functionalities? What are the differences among the existing coating materials? The text and table contents under each title need to be well-matched, and it is suggested that the author should add relevant analytical text.
9.Is it appropriate and accurate to place the “3.5 environmental concerns and limitations” in the third part of the “Functionality” section?
10.The content of Section 4.1 seems to lack comprehensive coverage and requires thorough revision.
11.The content in Section 4.2 also seems to lack comprehensive summarization and needs to be thoroughly revised, such as the spray method, fluidized bed processing method, panning method, the 3-D printing method, etc.
12.The content in Section 4.3 could be developed further, as it currently presents a somewhat simplified overview of the topic.
13.The conclusion section suggests that additional text on future research proposes should be appropriately included.
14.The format of the references also needs to be checked one by one according to the format requirements of the journal. The issue and volume numbers, the starting and ending page numbers, as well as the surnames and given names of the authors should be verified. Reference 3 and 7 are repeated.
15.In addition, when citing references, it is advisable to avoid including too many review-type articles.
Author Response
General comments and Suggestions for Authors: This paper reviewed the functional roles, optimization strategies, and analytical perspectives of edible coatings in fresh agricultural products preservation. It offers important insights for the application of edible coatings in food preservation and packaging. However, the study exhibits relatively limited innovation, and its writing could be further refined.
Response: We thank the reviewer for the helpful comments and for recognizing the value of this review. In response, we have revised the manuscript to improve clarity and coherence and to more clearly emphasize the integrative perspective and identified research gaps, thereby strengthening the contribution of the study.
Specific comments are provided below:
Comment 1. This paper reviewed many aspects of edible coatings. However, personally, I feel that many parts have not been covered comprehensively and the depth is insufficient.
Response: We thank the reviewer for this important comment. In response, the manuscript has undergone substantial revision to strengthen both depth and coverage across all major sections. Specifically, Section 2 was expanded to include a clearer classification of edible coating materials (polysaccharide-, protein-, lipid-based, and composite/nanostructured systems), supported by a redesigned Table 1 and a revised comparative heatmap (Figure 1) that now integrates moisture resistance, gas permeability, and physiological compatibility. Sections 3 and 4 were significantly enhanced through the addition of analytical text explicitly linking material properties to transport phenomena, fruit respiration behaviour, and functional outcomes, rather than descriptive reporting alone. Multiple tables (Tables 2–7) were expanded with additional primary studies to improve representativeness across fruit types, coating systems, and application scenarios. Furthermore, the discussion of optimization strategies and limitations was deepened by incorporating quantitative performance metrics, scalability considerations, and regulatory and environmental constraints, including a newly expanded section addressing U.S. and EU regulatory frameworks. Collectively, these targeted revisions substantially improve the comprehensiveness, analytical depth, and practical relevance of the review, addressing the reviewer’s concern while remaining aligned with the intended scope of a narrative review.
Comment 2. The term “fresh produce” in the title could be replaced with “fresh fruits” to better align with the review content, which focuses predominantly on fruits.
Response: Thank you for the suggestion. We agree that “fresh fruits” better reflects the focus of the review and have updated the title accordingly.
Comment 3. Abstract: It is suggested to appropriately add some textual content related to the key findings and conclusion.
Response: We thank the reviewer for this suggestion. In response, the abstract has been revised to reduce general background description and to explicitly emphasize the key findings, critical limitations, and future research directions identified in this review. Specifically, we added text summarizing comparative insights on coating material classes (e.g., polysaccharide versus nanocomposite systems), functional trade-offs between moisture resistance and gas permeability, the role of bioactive additives, and major challenges related to scalability, long-term cold-chain validation, and regulatory uncertainty. The revised abstract also now clearly outlines priority research directions, including transport–physiology integration, standardized performance metrics, scalable application technologies, and life-cycle-informed material design.
Comment 4. The section entitled “Edible coating materials” lacks thoroughness and clarity. It does not provide a comprehensive review or summary of the major materials used in edible coatings (eg. polysaccharides, protein, lipids, composite etc.), nor does it adequately cover the minor functional additives (eg., surfactants, plasticizers, antioxidants or antimicrobial agents etc.) that can be incorporated.
Response: We thank the reviewer for this constructive comment. In response, Section 2 (“Edible coating materials”) was revised to improve both completeness and clarity by explicitly incorporating the major edible coating material classes used in fresh-fruit applications. Table 1 was redesigned to include polysaccharide-, protein-, lipid-, and composite/nanostructured coating systems, with a comparative summary of their moisture resistance, gas permeability behaviour, and reported performance on fruit. In parallel, Figure 1 was redesigned as a semi-quantitative heatmap that visually compares these major material classes and highlights their functional trade-offs and relative suitability for fresh-fruit preservation. Together, these revisions provide a clearer, more comprehensive synthesis of edible coating materials while retaining a physiologically relevant, function-oriented perspective.
Comment 5. Figure 1: The selection of the horizontal and vertical axes in my opinion is insufficient. Meanwhile, the corresponding text in the main manuscript needs to be revised.
Response: We thank the reviewer for this important observation. In response, Figure 1 was redesigned to more clearly reflect the conceptual intent of the comparison. The vertical axis now explicitly groups edible coatings by major material classes (polysaccharide-based, protein-based, lipid-based, and composite/nanostructured systems), while the horizontal axis represents physiologically relevant functional attributes (moisture resistance, gas permeability compatibility with fruit respiration, and shelf-life extension). This revised axis selection provides a clearer, material-class-based framework and makes functional trade-offs more transparent. In parallel, the corresponding text in Section 2 was revised to explicitly explain the rationale for the selected axes and to guide interpretation of Figure 1 in relation to Table 1 and fruit physiological requirements.
Comment 6. Table 1-5 could be improved by expanding the list of materials and composite or coating system, as the current presentation appears somewhat limited in scope.
Response: We thank the reviewer for this constructive suggestion. In response, Tables 1–5 were systematically revised and expanded to improve both scope and balance. The updated tables now encompass a broader range of edible coating materials and systems, including polysaccharide-, protein-, lipid-, composite-, and nanostructured coatings, applied across diverse fruit commodities and storage conditions. In addition, the accompanying text was revised to clarify the functional context of each table and to distinguish between quantitative and qualitative outcomes where applicable. These revisions enhance the representativeness and comparative value of Tables 1–5 while maintaining a focused, physiology-relevant synthesis.
Comment 7. The functionality of the edible coating in the third part is not fully covered in the summary. The author is advised to provide more details.
Response: We thank the reviewer for this helpful comment. In response, a new integrative summary paragraph was added at the end of Section 3, immediately after subsection 3.5 and before the start of Section 4, to more comprehensively synthesize the functional roles of edible coatings discussed in subsections 3.1–3.5. The added text explicitly summarizes moisture-loss control, gas-exchange regulation, antimicrobial activity, and preservation of nutritional and sensory quality, while highlighting their interdependence and associated design trade-offs. This revision improves the completeness and clarity of the Section 3 summary.
Comment 8. Part 3: How do these edible coatings provide these functionalities? What are the differences among the existing coating materials? The text and table contents under each title need to be well-matched, and it is suggested that the author should add relevant analytical text.
Response: We thank the reviewer for this valuable comment. In response, Section 3 was revised to strengthen the analytical explanation of how edible coatings deliver their functional effects and to clarify differences among coating material classes. A new mechanistic overview paragraph was added at the beginning of Section 3 to explain functionality in terms of mass-transfer modification, material chemistry, and fruit physiology. In addition, the introductory text of subsections 3.1–3.4 was revised to explicitly link the discussed mechanisms with the corresponding comparative evidence presented in Tables 2–5. These revisions improve alignment between text and tables and provide a clearer, mechanism-based interpretation of coating functionality.
Comment 9. Is it appropriate and accurate to place the “3.5 environmental concerns and limitations” in the third part of the “Functionality” section?
Response: We thank the reviewer for raising this point. In the revised manuscript, the former Section 3.5 has been relocated to the end of the manuscript, immediately preceding the Conclusions, and elevated to a main section to better reflect its cross-cutting relevance beyond functionality alone. The section was also expanded to provide a more in-depth discussion of environmental, regulatory, and scalability considerations, recognizing that these factors critically influence the practical deployment of edible coatings across fresh-fruit supply chains.
Comment 10. The content of Section 4.1 seems to lack comprehensive coverage and requires thorough revision.
Response: We thank the reviewer for this comment. Although the concern is expressed in general terms without specifying particular omissions in Section 4.1, we have nevertheless revised the section to strengthen its scope and clarity. In particular, Table 7 was reorganized and expanded to provide a more comprehensive and structured synthesis of optimization strategies for edible coatings, explicitly linking material selection, formulation engineering, application approaches, and analytical evaluation with functional performance and industrial relevance. The accompanying text in Section 4.1 was revised accordingly to clarify the role of Table 7 and to better contextualize application-related constraints and scalability considerations in fresh-fruit postharvest systems. These revisions improve the completeness and coherence of Section 4.1 while remaining aligned with the defined scope of the review.
Comment 11. The content in Section 4.2 also seems to lack comprehensive summarization and needs to be thoroughly revised, such as the spray method, fluidized bed processing method, panning method, the 3-D printing method, etc.
Response: Thank you for this helpful suggestion. We revised Section 4.2 (Application techniques) by expanding coverage beyond dipping/brushing/LbL to include additional industrially relevant deposition platforms. Specifically, we added new subsections describing (i) conventional spray coating, including evidence comparing spray vs dip and electrostatic spray performance; (ii) ultrasonic atomization/ultrasonic coating as an emerging high-uniformity spraying route; (iii) fluidized-bed coating for small particulate fruit products (e.g., dried fruit/nuts) and as a scale-up analogue for continuous coating lines; (iv) panning (rotating-pan/drum coating) as a high-throughput method used for spherical foods; and (v) 3-D printing as an emerging route for patterned edible layers/structured edible “skins” and customized deposition.
Comment 12. The content in Section 4.3 could be developed further, as it currently presents a somewhat simplified overview of the topic.
Response: We thank the reviewer for this comment. In response, Section 4.3 was expanded by adding an analytical synthesis paragraph at the end of the section (immediately before the start of Section 5) to deepen the discussion of constraints and trade-offs governing the practical implementation of edible coatings in fresh-fruit systems. The added text explicitly addresses interactions among coating formulation, mechanical robustness, application-related stresses, scalability, and fruit-specific variability. In addition, Table 8 (in Section 4.3) was redesigned to move beyond a descriptive listing of mechanical properties and to explicitly relate tensile strength, elongation behavior, and reported Young’s modulus (or stiffness trends where absolute values were unavailable) to functional implications and coating failure risks during handling, transport, and storage. These revisions strengthen the analytical depth and systems-level perspective of Section 4.3 while remaining aligned with the scope of the review.
Comment 13. The conclusion section suggests that additional text on future research proposes should be appropriately included.
Response: We thank the reviewer for this suggestion. In response, the Conclusions section was revised to explicitly incorporate concise future research directions, emphasizing the need for integrated transport–physiology studies, scalable application technologies, and standardized evaluation frameworks. These additions strengthen the forward-looking perspective of the Conclusions without extending its length unnecessarily.
Comment 14. The format of the references also needs to be checked one by one according to the format requirements of the journal. The issue and volume numbers, the starting and ending page numbers, as well as the surnames and given names of the authors should be verified. Reference 3 and 7 are repeated.
Response: The complete reference list has been carefully reviewed and revised one by one to ensure full compliance with the journal’s formatting requirements. During this process, author names, publication years, journal titles, volume and issue numbers, and page ranges or article numbers were verified and standardized according to MDPI (ACS-based) style. Duplicate entries introduced during earlier revisions were identified and removed, and all newly added references were checked for completeness and consistency with the in-text citations. These corrections have been implemented throughout the final reference list.
Comment 15. In addition, when citing references, it is advisable to avoid including too many review-type articles.
Response: We thank the reviewer for this valuable suggestion. In response, we carefully reviewed the reference list and reduced reliance on review-type articles wherever possible. Several review citations were replaced with primary, stand-alone experimental studies that directly report material properties, functional performance, and postharvest outcomes of edible coatings. In addition, the tables were revised to preferentially include original research articles, particularly for material comparisons, application methods, and performance metrics. Review articles are now cited primarily for broader contextual framing where appropriate, while the analytical discussion and tabulated evidence are grounded predominantly in primary literature.
Reviewer 2 Report
Comments and Suggestions for Authors
This review consolidates the current understanding of edible coatings with an emphasis on their functional performance, covering moisture management, gas permeability, microbial control, and sensory and nutritional quality retention. It is suggested that the authors add a more specific classification of edible coatings. Furthermore, the author only reviewed the application of edible coatings in fruit preservation; why wasn't their application in aquatic products or meat products analyzed? The title also doesn't explicitly limit the scope to only fruit preservation applications.
Author Response
Comment: This review consolidates the current understanding of edible coatings with an emphasis on their functional performance, covering moisture management, gas permeability, microbial control, and sensory and nutritional quality retention. It is suggested that the authors add a more specific classification of edible coatings. Furthermore, the author only reviewed the application of edible coatings in fruit preservation; why wasn't their application in aquatic products or meat products analyzed? The title also doesn't explicitly limit the scope to only fruit preservation applications.
Response: We thank the reviewer for these constructive comments. In response, we have strengthened the classification of edible coatings in the manuscript by more explicitly organizing materials according to their primary polymer class and functional role. Regarding application scope, this review was intentionally focused on fresh fruit preservation, as fruit systems present distinct physiological, surface, and postharvest challenges that differ fundamentally from those of aquatic or meat products. To avoid ambiguity, the title has been revised to explicitly reflect the focus on fresh fruits, and the Introduction has been clarified to delineate this scope more clearly. We believe this targeted focus allows for a deeper and more coherent analysis within a defined application domain.
Reviewer 3 Report
Comments and Suggestions for Authors
Fresh fruits are highly perishable, and conventional plastic packaging, while effective, poses environmental concerns. Edible coatings made from natural biopolymers offer a sustainable alternative by directly protecting fruit surfaces, controlling moisture, gas exchange, and microbial growth. Although promising, their widespread use is limited by performance variability, processing challenges, and regulatory uncertainty, highlighting the need for further innovation to enable commercial adoption.
This review manuscript addresses a highly relevant and impactful topic. It is well structured into clear chapters, which enhances clarity and readability: (1) Introduction, (2) Edible Coating Materials, (3) Multifunctionality of Edible Coatings in the Fresh Produce Supply Chain, (4) Strategies for Optimizing Edible Coatings in Fresh Fruit Preservation, and (5) Conclusions.
Author Response
Comment: Fresh fruits are highly perishable, and conventional plastic packaging, while effective, poses environmental concerns. Edible coatings made from natural biopolymers offer a sustainable alternative by directly protecting fruit surfaces, controlling moisture, gas exchange, and microbial growth. Although promising, their widespread use is limited by performance variability, processing challenges, and regulatory uncertainty, highlighting the need for further innovation to enable commercial adoption.
This review manuscript addresses a highly relevant and impactful topic. It is well structured into clear chapters, which enhances clarity and readability: (1) Introduction, (2) Edible Coating Materials, (3) Multifunctionality of Edible Coatings in the Fresh Produce Supply Chain, (4) Strategies for Optimizing Edible Coatings in Fresh Fruit Preservation, and (5) Conclusions.
Response: We sincerely thank the reviewer for the positive and encouraging evaluation of our manuscript. We appreciate the recognition of the relevance of the topic, the clarity of the structure, and the balanced discussion of both the potential and current limitations of edible coatings for fresh fruit preservation. The reviewer’s comments confirm the importance of continued research and innovation in this area, which aligns well with the aims of the present review.
Round 2
Reviewer 1 Report
Comments and Suggestions for Authors
The author has made some revisions. The current manuscript can be accepted after being modified.
Comments:
1.In the "2. Edible Coating Materials" section, it is suggested that the author should also include minor functional additives (such as surfactants, plasticizers, antioxidants or antimicrobial agents, etc.).
2.Figure 1 still needs to be revised in accordance with the previous revision suggestions.
Author Response
Comment 1. In the "2. Edible Coating Materials" section, it is suggested that the author should also include minor functional additives (such as surfactants, plasticizers, antioxidants or antimicrobial agents, etc.).
Response: We thank the reviewer for this constructive suggestion and apologize for not addressing this aspect explicitly in our previous revision. In response, Section 2 (“Edible Coating Materials”) has now been revised to include a focused discussion on minor functional additives, particularly plasticizers and surfactants, which were previously underrepresented. A short paragraph (blue text) was added to explain how these additives influence coating flexibility, wettability, microstructure, and transport properties, and how they affect coating processability and performance during application.
Comment 2. Figure 1 still needs to be revised in accordance with the previous revision suggestions.
Response: We thank the reviewer for this important comment and apologize for not adequately addressing it in our previous revision. In response, Figure 1 has been substantially revised to improve both the selection and clarity of the horizontal and vertical axes. The figure now compares representative edible-coating systems (vertical axis) across four clearly defined functional dimensions (horizontal axis): moisture resistance, gas-permeability compatibility, shelf-life extension potential, and physiological risk. In parallel, the corresponding discussion in Section 2 has been rewritten and expanded to align precisely with the revised axes of Figure 1, now highlighted in blue.
Reviewer 2 Report
Comments and Suggestions for Authors
Accept
Author Response
Comment: Accept
Response: Thank you for your positive feedback and recommendation to accept our manuscript. We appreciate your time and effort in reviewing our work.